# The human disease-associated gene *ZNFX1* controls inflammation through inhibition of the NLRP3 inflammasome

Jing Huang[1,5], Yao Wang [2,5✉], Xin Jia[2], Changfeng Zhao [1], Meiqi Zhang[2], Mi Bao[1], Pan Fu[1], Cuiqin Cheng [2], Ruona Shi[3,4], Xiaofei Zhang [3,4], Jun Cui [1], Gang Wan [1✉] & Anlong Xu [1,2✉]

## Abstract

**Inherited deficiency of zinc finger NFX1-type containing 1 (ZNFX1), a dsRNA virus sensor, is associated with severe familial immunodeficiency, multisystem inflammatory disease, increased susceptibility to viruses, and early mortality. However, limited treatments for patients with pathological variants of ZNFX1 exist due to an incomplete understanding of the diseases resulting from ZNFX1 mutations. Here, we demonstrate that ZNFX1 specifically inhibits the activation of the NLR family pyrin domain-containing protein 3 (NLRP3) inflammasome in response to NLRP3 activators both in vitro and in vivo. ZNFX1 retains NLRP3 in the cytoplasm and prevents its accumulation in the TGN38 + /TGN46+ vesicles in the resting state. Upon NLRP3 inflammasome activation, ZNFX1 is cleaved by caspase-1, establishing a feed-forward loop that promotes NLRP3 accumulation in the trans-Golgi network (TGN) and amplifies the activity of the downstream cascade. Expression of wild-type ZNFX1, but not of ZNFX1 with human pathogenic mutations, rescues the impairment of NLRP3 inflammasome inhibition. Our findings reveal a dual role of ZNFX1 in virus sensing and suppression of inflammation, which may become valuable for the development of treatments for ZNFX1 mutation-related diseases.**

**Keywords** ZNFX1; NLRP3 Inflammasome; Hyperinflammation; Feed-Forward Loop; Genetic Disease
**Subject Categories** Immunology; Microbiology, Virology & Host Pathogen Interaction

## Introduction

The inflammasome is a multiprotein complex that functions as a platform for the caspase-1-dependent activation of proinflammatory cytokines and the induction of pyroptosis, a type of programmed inflammatory cell death (Bergsbaken et al, 2009; Swanson et al, 2019). Among the various inflammasome complexes, the NLR family pyrin domain-containing protein 3 (NLRP3) inflammasome plays a crucial role in the host immune defense against bacterial, fungal, and viral infections (Allen et al, 2009; Kanneganti et al, 2006; Nozaki et al, 2022; Thomas et al, 2009).

The activation of the NLRP3 inflammasome is initiated by the accumulation of NLRP3 oligomers into cage-like structures on TGN38+/TGN46+ membrane vesicles that may be part of the trans-Golgi network (TGN). This network is initially in an inactive, self-inhibitory state, poised for further activation (Andreeva et al, 2021; Xiao et al, 2023). In response to various stimuli, dispersed TGN38+/ TGN46+ vesicles form, and oligomeric NLRP3 interacts with these dispersed vesicles through the polybasic region of NLRP3 and PI4P within these vesicles. Originally, these dispersed vesicles were believed to constitute a dispersed trans-Golgi network (dTGN) (Chen and Chen, 2018). However, recent studies have revealed that these dispersed vesicles are endosomal vesicles that fail to be retrogradely transported to the TGN (Lee et al, 2023; Zhang et al, 2023). Oligomeric NLRP3 on these dispersed vesicles then recruits the adapter protein ASC, which undergoes prion-like aggregation, forming a single large speck in the perinuclear region (Dick et al, 2016; Lu et al, 2014). The mature inflammasome is formed when ASC binds to caspase-1 (Bryan et al, 2009). Subsequently, caspase-1 is cleaved and activated, leading to the maturation and secretion of the proinflammatory cytokines IL-1β and IL-18 (Broz and Dixit, 2016; Dinarello, 2018; Liu and Cao, 2016).

While NLRP3 is vital for initiating inflammation and facilitating the immune response, excessive and prolonged activation of the NLRP3 inflammasome can have detrimental effects and contribute to the development of various diseases, including diabetes, atherosclerosis, and gout (Broz and Dixit, 2016; Guo et al, 2015;

[1]Guangdong Provincial Key Laboratory of Pharmaceutical Functional Genes, MOE Key Laboratory of Gene Function and Regulation, State Key Laboratory of Biocontrol, School of Life Sciences, Sun Yat-sen University, Guangzhou, Guangdong 510275, China. [2]Beijing Research Institute of Chinese Medicine, School of Life Science, Beijing University of Chinese Medicine, Beijing 100029, China. [3]CAS Key Laboratory of Regenerative Biology, Guangdong Provincial Key Laboratory of Stem Cell and Regenerative Medicine, GIBH-HKU Guangdong-Hong Kong Stem Cell and Regenerative Medicine Research Centre, Guangzhou Institutes of Biomedicine and Health, Chinese Academy of Sciences, Guangzhou, Guangdong 510530, China. [4]Center for Cell Lineage and Atlas, BioLand Laboratory, Guangzhou Regenerative Medicine and Health Guangdong Laboratory, Guangzhou, Guangdong 510530, China. [5]These authors contributed equally: Jing Huang, Yao Wang. ✉E-mail: yaowang@bucm.edu.cn; wangang5@mail.sysu.edu.cn; lssxal@mail.sysu.edu.cn

Liu and Cao, 2016). Therefore, precise control of NLRP3 inflammasome activation is crucial for maintaining immune homeostasis and preventing the onset of inflammatory disorders.

Zinc finger NFX1-type containing 1 (ZNFX1) is a highly conserved RNA helicase belonging to the SF1 helicase family. Previously, we identified ZNFX1 as an interferon-stimulated gene (ISG) and a sensor for double-stranded RNA (dsRNA) viruses. It plays a crucial role in initiating the production of IFNs and the expression of ISGs during the early stages of viral infection (Wang et al, 2019). Recent clinical studies conducted by different research groups have highlighted the association of loss-of-function mutations in ZNFX1 with familial multisystem inflammatory disease and severe viral infections (Al-Saud et al, 2023; Alawbathani et al, 2022; Le Voyer et al, 2021; Vavassori et al, 2021). This suggests previously unrecognized roles for ZNFX1 in these pathogenic processes.

Here we demonstrate that ZNFX1 specifically inhibits NLRP3 inflammasome activation by sequestering and directly interacting with NLRP3 in the cytoplasm during the priming stage. Furthermore, we found that the cleavage of ZNFX1 by caspase-1 during the activation stage increases NLRP3 inflammasome activation. These findings identify ZNFX1 as a key regulator that coordinates viral sensing and NLRP3 inflammasome activation, ultimately influencing the fate of immune cells during pathogen invasion.

## Results

### ZNFX1 plays a specific role in inhibiting the activation of the NLRP3 inflammasome in vitro

To investigate the connection between ZNFX1 deficiency and the inflammasome, we utilized the CRISPR/Cas9 gene editing system to knock out ZNFX1 (ZNFX1 KO) in immortalized mouse bone marrow-derived macrophages (iBMDMs) and THP-1 human monocytic cells. Upon exposure of ZNFX1 KO macrophages to the NLRP3 activator nigericin, we observed a 2 to 5-fold increase in the production of IL-1β and cleaved caspase-1 (Figs. 1A–D and EV1A–K; Appendix Fig. S1A–E). Importantly, this effect was not observed when macrophages were exposed to agonists such as poly(dA:dT), MDP, and Salmonella, which activate the AIM2, NLRP1b, or NLRC4 inflammasome, respectively (Figs. 1A,C and EV1A–E,H,I; Appendix Fig. S1A,B,D). To validate the specificity of ZNFX1 in inhibiting the NLRP3 inflammasome, we stimulated both wild-type (WT) and ZNFX1 KO macrophages with various other NLRP3 agonists, and increased levels (1.5- to 5-fold) of IL-1β secretion and caspase-1 cleavage were observed in ZNFX1 KO macrophages (Figs. 1A–D and EV1A–K; Appendix Figs. S1A–F and S2A–F). Notably, the ability of all the tested stimuli to induce TNF-α release was unaffected in ZNFX1 KO macrophages (Appendix Fig. S2G,H).

To further confirm the suppressive effect of ZNFX1 on NLRP3 inflammasome activation, we generated ZNFX1 rescue (ZNFX1 RE) and ZNFX1 overexpression (ZNFX1 OE) cell lines by reintroducing full-length ZNFX1 into ZNFX1 KO THP-1 or iBMDM cells. We evaluated the formation of cytosolic ASC aggregates, known as ASC specks, after THP-1 cells were stimulated with the NLRP3 activator nigericin (Fig. 1E,F). ASC speck formation was approximately

threefold greater in ZNFX1 KO cells than in WT cells, but it returned to WT levels in ZNFX1 RE cells and was significantly reduced in ZNFX1 OE cells (Fig. 1E,F). Conversely, no significant differences were observed following stimulation with poly (dA:dT) (Fig. 1E,F). Immunoblot analysis further demonstrated that ASC oligomerization was increased in ZNFX1 KO cells, restored in ZNFX1 RE cells, and significantly diminished in ZNFX1 OE cells compared to WT cells (Figs. 1G,H and EV2A,B; Appendix Fig. S1G,I). Moreover, using an α-NLRP3 antibody, we detected a large oligomeric complex containing NLRP3 upon exposure of macrophages to nigericin, indicating NLRP3 inflammasome assembly (Fig. EV2D). NLRP3 oligomerization was greater in ZNFX1 KO cells, similar to that in ZNFX1 RE cells, and significantly decreased in ZNFX1 OE cells compared to WT cells (Fig. EV2D). Given that the NLRP3 protein level is a key limiting factor for inflammasome activation (Dowling and O'Neill, 2012; Song et al, 2016), we investigated whether ZNFX1 affects NLRP3 protein levels. Immunoblotting showed that NLRP3 protein levels were similar regardless of the ZNFX1 expression level (Fig. EV2E–H). Additionally, we assessed the levels of cleaved caspase-1 and IL-1β and observed similar patterns (Figs. 1G,H and EV2A,B,D; Appendix Fig. S1G–J).

To further validate our results, we obtained primary BMDMs from WT and $Znfx1^{-/-}$ mice and confirmed that NLRP3 inflammasome activation was increased (1.5- to 2.5-fold) in $Znfx1^{-/-}$ BMDMs upon NLRP3 agonist treatment but not upon NLRC4 or AIM2 agonist treatment (Figs. 1I–K and EV2I–N; Appendix Fig. S2I). Collectively, these findings suggest that ZNFX1 specifically suppresses NLRP3 inflammasome activation in human and mouse macrophages.

### ZNFX1 suppresses the activation of the NLRP3 inflammasome in vivo

Having established the role of ZNFX1 in suppressing the activation of the NLRP3 inflammasome at the cellular level, we next sought to investigate its effects in mouse models. To this end, we intraperitoneally injected LPS, which induces acute inflammation and can be lethal, into WT and $Znfx1^{-/-}$ mice. We found that while WT mice survived for 6 days post-LPS treatment in our experimental setting (when the experiments were stopped), all $Znfx1^{-/-}$ mice died within 4 days. This suggests that ZNFX1 protects mice from lethal inflammation (Fig. 2A,B). The induction of IL-1β expression by intraperitoneal injection of LPS is known to be NLRP3 dependent (Mariathasan et al, 2006; Martinon et al, 2006), and we observed a significantly greater concentration of IL-1β in the serum of $Znfx1^{-/-}$ mice than in that of WT mice (Fig. 2C). When mice were administered MCC950, a specific inhibitor of NLRP3 (Coll et al, 2019; Coll et al, 2015; Guo et al, 2022; Ismael et al, 2018; Schuh et al, 2019; Tapia-Abellán et al, 2019), IL-1β secretion was completely blocked in both WT and $Znfx1^{-/-}$ mice (Fig. 2C,D).

The experimental autoimmune encephalomyelitis (EAE) mouse model is a widely used model of the human disease multiple sclerosis (Braga et al, 2019; Gris et al, 2010; Hou et al, 2020). Previous research has demonstrated that the development of EAE requires NLRP3 (Gris et al, 2010). $Znfx1^{-/-}$ mice exhibited greater clinical scores than did WT mice (Fig. 2E), and they experienced more severe weight loss than did WT mice after EAE induction

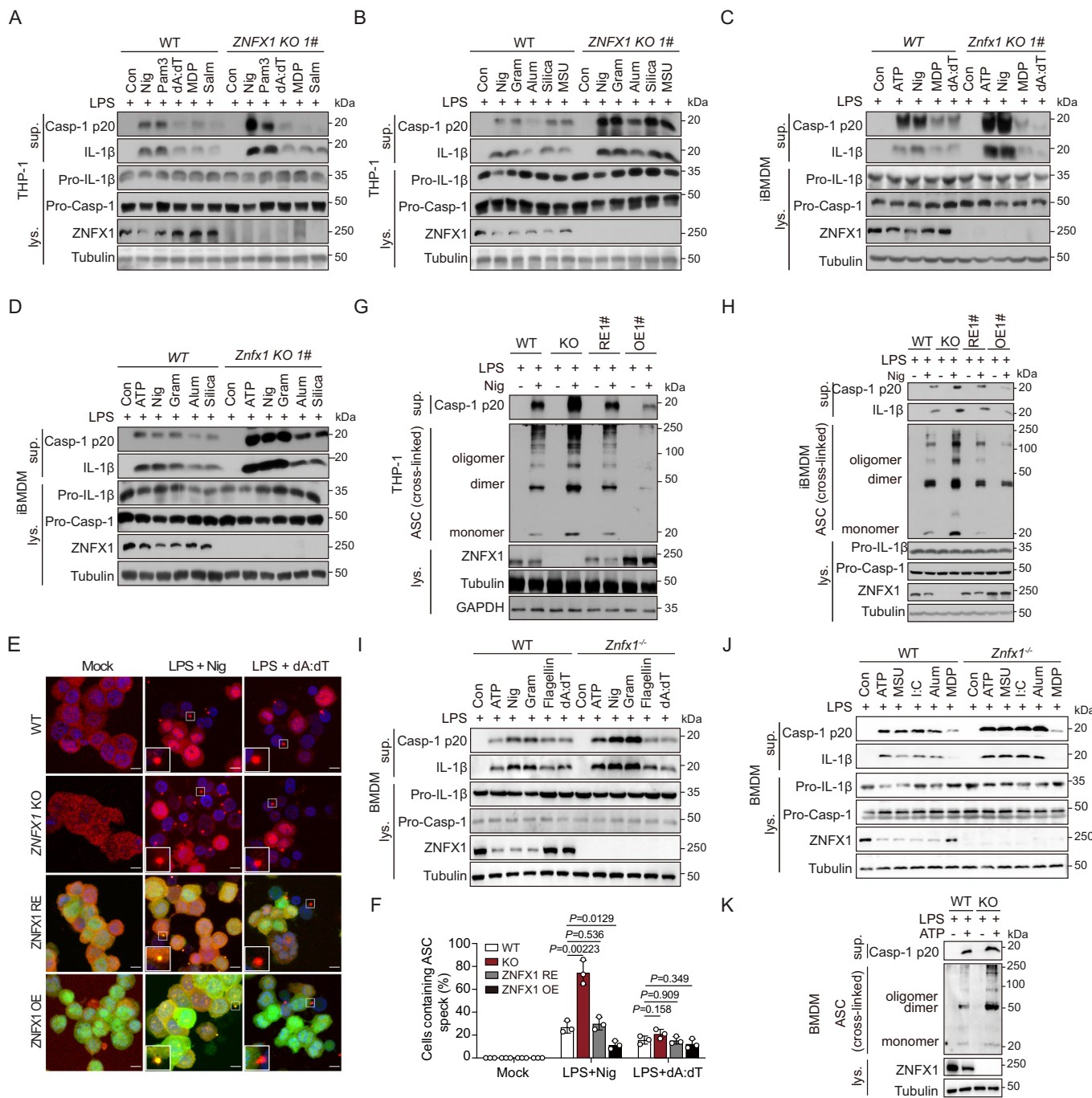

**Figure 1. ZNFX1 specifically suppresses the activation of the NLRP3 inflammasome in vitro.**

(A–D) LPS-primed wildtype (WT) or *ZNFX1* knockout (KO) THP-1 derived macrophages (A, B) and *Znfx1* KO iBMDM (C, D) were treated with the indicated inflammation agonists, caspase-1 and IL-1β in the supernatant (sup.) and cell lysate (Lys.) were separated by SDS-PAGE and immunoblotted with the indicated antibodies. Con control, Nig nigericin, Pam3 Pam3CSK4, Salm salmonella, Gram gramicidin. (E, F) A cassette expressing WT *ZNFX1* was reintroduced to *ZNFX1* KO THP-1 to generate *ZNFX1* rescue cell lines (hereafter referred to as *ZNFX1* rescue, with *ZNFX1* expression levels similar to WT parental cells) and *ZNFX1* overexpressing cells (hereafter referred to as *ZNFX1* OE, with *ZNFX1* expression levels higher than WT parental cells). Primed WT, *ZNFX1* KO, *ZNFX1* rescue, and *ZNFX1* OE THP-1-derived macrophages were treated with ethanol, nigericin, or poly (dA:dT). ASC specks were detected with an anti-ASC antibody followed by Alex fluor-568 conjugated secondary antibody (E). Scale bar 10 μm. The bottom left magnified area indicates ASC speck. The percentage of cells with ASC specks was quantified from 100 cells (F). n = 3 biological replicates, mean ± s.d., two-sided student's *t*-test. (G, H) Primed WT, *ZNFX1* KO, *ZNFX1* rescue, and *ZNFX1* OE THP-1-derived macrophages or iBMDM cells were treated with ethanol and nigericin. Immunoblot analysis was conducted to examine caspase-1 activation in the supernatants and ASC oligomerization in the lysates of nigericin-treated macrophages. (I, J) LPS-primed WT or *Znfx1* KO primary BMDM cells were treated with the indicated inflammation agonists. Proteins in the supernatant and cell lysate were separated by SDS-PAGE and immunoblotted with the indicated antibodies. (K) Immunoblot analysis of caspase-1 activation in the supernatants and ASC oligomerization in the lysates of LPS + ATP-treated primary WT and *Znfx1* KO BMDM cells. Source data are available online for this figure.

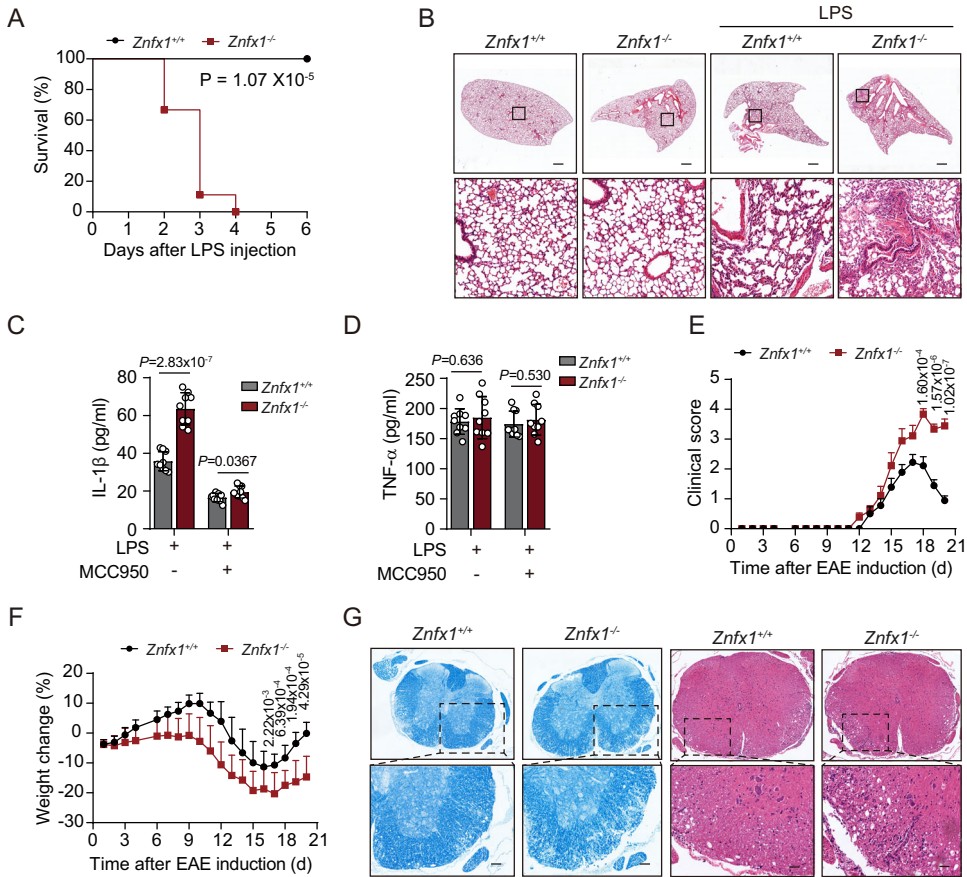

**Figure 2. ZNFX1 inhibits the activation of the NLRP3 inflammasome in vivo.**

(A) WT and *Znfx1⁻/⁻* mice were injected with 30 mg/kg LPS, and the mortality of the mice was recorded daily post-injection. $n = 9$ mice for each genotype. Log-rank (Mantel-Cox) test. (B) WT and *Znfx1⁻/⁻* mice were injected with 30 mg/kg LPS. Lung structure was examined using pathological sections. Scale bar, 100 μm. (C, D) WT and *Znfx1⁻/⁻* mice were injected with 20 mg/kg LPS and fed with or without 50 mg/kg MCC950. The protein levels of secreted IL-1β (C) and TNF-α (D) were measured by ELISA. $n = 9$ mice for each genotype, mean ± s.d., Student's *t*-test, two-tailed. (E) Clinical score after EAE induction of WT and *Znfx1⁻/⁻* mice. $n = 9$ mice for each genotype, mean ± s.d., Student's *t*-test, two-tailed. (F) The weight of WT and *Znfx1⁻/⁻* mice was measured daily after EAE induction. $n = 9$ mice for each genotype, mean ± s.e.m., Student's *t*-test, two-tailed. (G) Spinal Cords of WT and *Znfx1⁻/⁻* mice were stained by luxol fast blue staining and hematoxylin-eosin staining after EAE induction. Scale bar, 100 μm. Source data are available online for this figure.

(Fig. 2F). These results suggest that the severity of EAE was worse in the *Znfx1⁻/⁻* mice than in the WT mice (Fig. 2G). Taken together, these observations indicate that ZNFX1 plays an inhibitory role in two mouse models of NLRP3-dependent inflammation, establishing the role of ZNFX1 in suppressing NLRP3 inflammation in vivo.

## ZNFX1 interacts with NLRP3

We aimed to investigate the mechanism by which ZNFX1 regulates the NLRP3 inflammasome and hypothesized that ZNFX1 interacts with components of the NLRP3/ASC/Caspase-1 complex. To test this hypothesis, we performed coimmunoprecipitation experiments, which revealed that ZNFX1 interacted with NLRP3 but not with caspase-1, ASC, or AIM2, another inflammasome receptor (Fig. 3A). Furthermore, when immunoprecipitation of endogenous NLRP3 was performed with an α-NLRP3 antibody in resting THP-1, iBMDM, or BMDM cell extracts, ZNFX1 was pulled down, confirming that ZNFX1 did not interact with NLRP3

merely because it was overexpressed (Fig. 3B–D). Notably, the ZNFX1-NLRP3 interaction was enhanced when iBMDMs or BMDMs, but not THP-1 cells, were primed with LPS (Fig. 3B±D). When the NLRP3 inflammasome was activated with nigericin, ZNFX1 still interacted with NLRP3, albeit at a decreased level (Fig. 3B–D).

To further verify that the interaction between ZNFX1 and NLRP3 was direct, surface plasmon resonance (SPR) experiments were conducted using purified 3xFLAG-NLRP3 and 3xFLAG-ZNFX1 proteins (Fig. EV3A). The results indicated a stable and likely high-affinity interaction between ZNFX1 and NLRP3 (Fig. EV3B,C). Confocal microscopy analysis revealed that ZNFX1 diffusely colocalized with NLRP3 in the cytoplasm of THP-1 and HeLa cells (Figs. 3E and EV3D). Proximity biotin labeling showed that biotinylated species catalyzed by miniturbo-ZNFX1 containing NLRP3, further confirming that ZNFX1 and NLRP3 are likely within 20 nm of each other in cells (Figs. 3F and EV3E,F). To estimate the stoichiometry of this interaction, we subjected THP-1 or iBMDM cell lysates to size-exclusion chromatography.

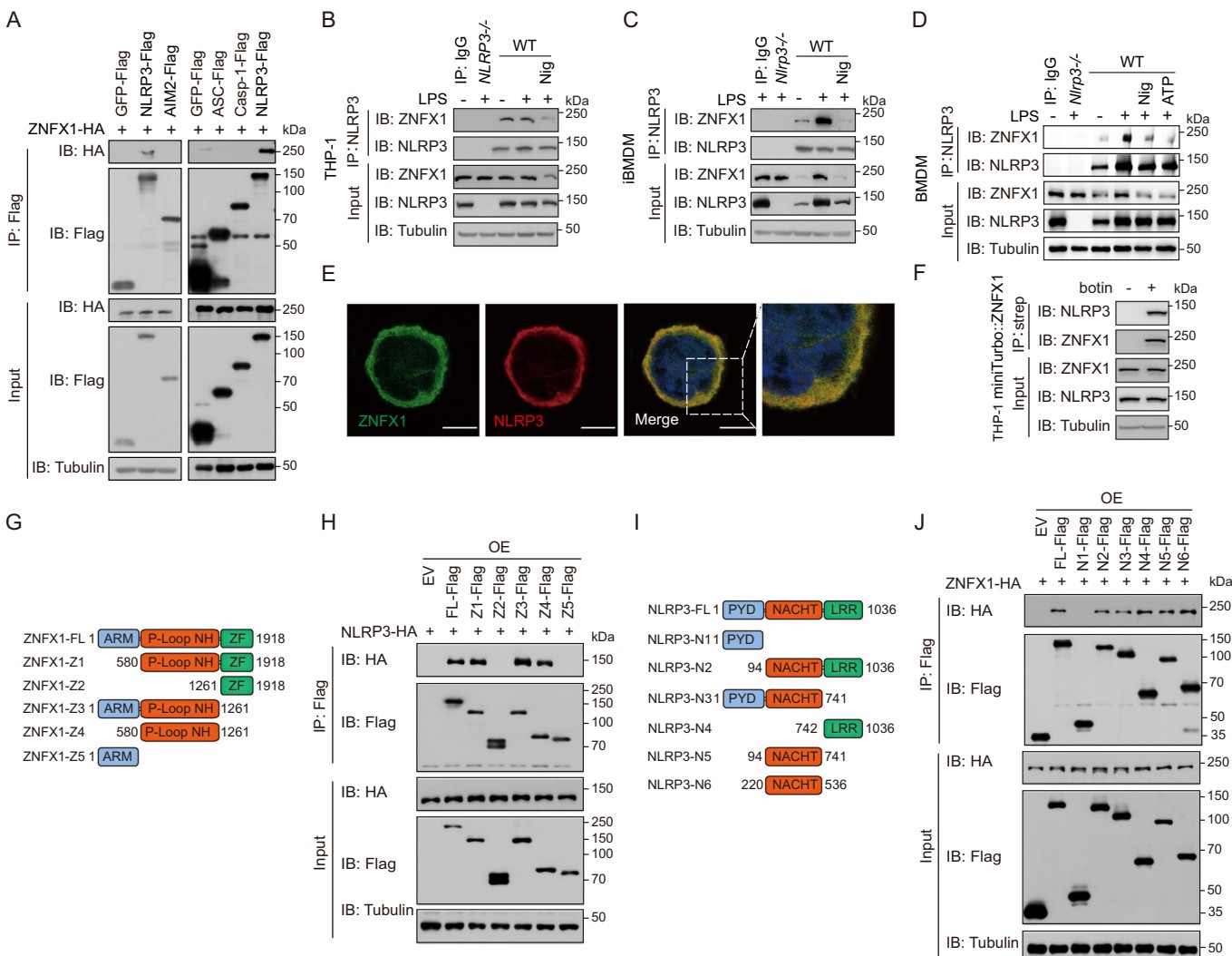

**Figure 3. ZNFX1 interacts with NLRP3.**

(A) FLAG-tagged proteins were immunoprecipitated with anti-FLAG-M2 beads and analyzed by immunoblotting. (B–D) Coimmunoprecipitation of endogenous ZNFX1 and NLRP3 from THP-1 derived macrophages, iBMDM, or primary BMDM using anti-NLRP3 antibody. (E) Colocalization between *ZNFX1* and NLRP3 was examined by immunofluorescence in THP-1-derived macrophages. Scale bar, 10 μm. (F) Immunoblotting of proteins in cell lysate and miniTurbo-ZNFX1 catalyzed biotinylated species from HeLa cells before and after adding biotin substrate with the indicated antibodies. (G) Schematic diagram of ZNFX1 and its truncation mutants. (H) FLAG-tagged full-length or different domains of ZNFX1 and HA-NLRP3 were co-expressed in HEK293T cells, and lysate was immunoprecipitated with anti-FLAG-M2 beads. Immunoprecipitated proteins or proteins from input lysate were analyzed with anti-FLAG, anti-HA, or anti-Tubulin antibodies. (I) Schematic diagram of NLRP3 and its truncation mutants. (J) FLAG-tagged full-length or different domains of NLRP3 and HA-ZNFX1 were co-expressed in HEK293T cells, and lysate was immunoprecipitated with anti-FLAG-M2 beads. Immunoprecipitated proteins or proteins from input lysate were analyzed with anti-FLAG, anti-HA, or anti-Tubulin antibodies. Similar results were obtained in three independent experiments. Source data are available online for this figure.

Immunoblotting analysis of elutions revealed that approximately 64–84% of NLRP3 species interacted with ZNFX1 (Fig. EV3G). Taken together, these results suggest that ZNFX1 likely directly interacts with the majority of NLRP3 in the cytoplasm in resting cells and that this interaction is dynamic during NLPR3 inflammasome priming and activation.

To identify the specific regions responsible for the ZNFX1-NLRP3 interaction, we conducted coimmunoprecipitation assays using a series of ZNFX1 and NLRP3 truncation mutants. The results indicated that full-length NLRP3 interacted with the P-loop domain of ZNFX1, which encompasses the conserved Walker A and Walker B motifs of the SF1 helicase (Fig. 3G,H). Conversely,

the full-length ZNFX1 interacted with the NACHT domain and LRR domain of NLRP3 (Fig. 3I,J). Since the NLRP3 activator NEK7 also interacts with the NACHT domain of NLRP3 (Shi et al, 2016), we investigated whether ZNFX1 competes with NEK7 for binding to NLRP3. Our coimmunoprecipitation analysis showed that the degree of interaction between NEK7 and NLRP3 was consistent regardless of whether ZNFX1 was deleted or overexpressed (Fig. EV3H, I). This observation suggests that NEK7 and ZNFX1 likely bind to different regions within the NACHT domain of NLRP3. Therefore, we concluded that ZNFX1 directly interacts with NLRP3 via the P-loop domain of ZNFX1 and the NACHT/LRR domain of NLRP3.

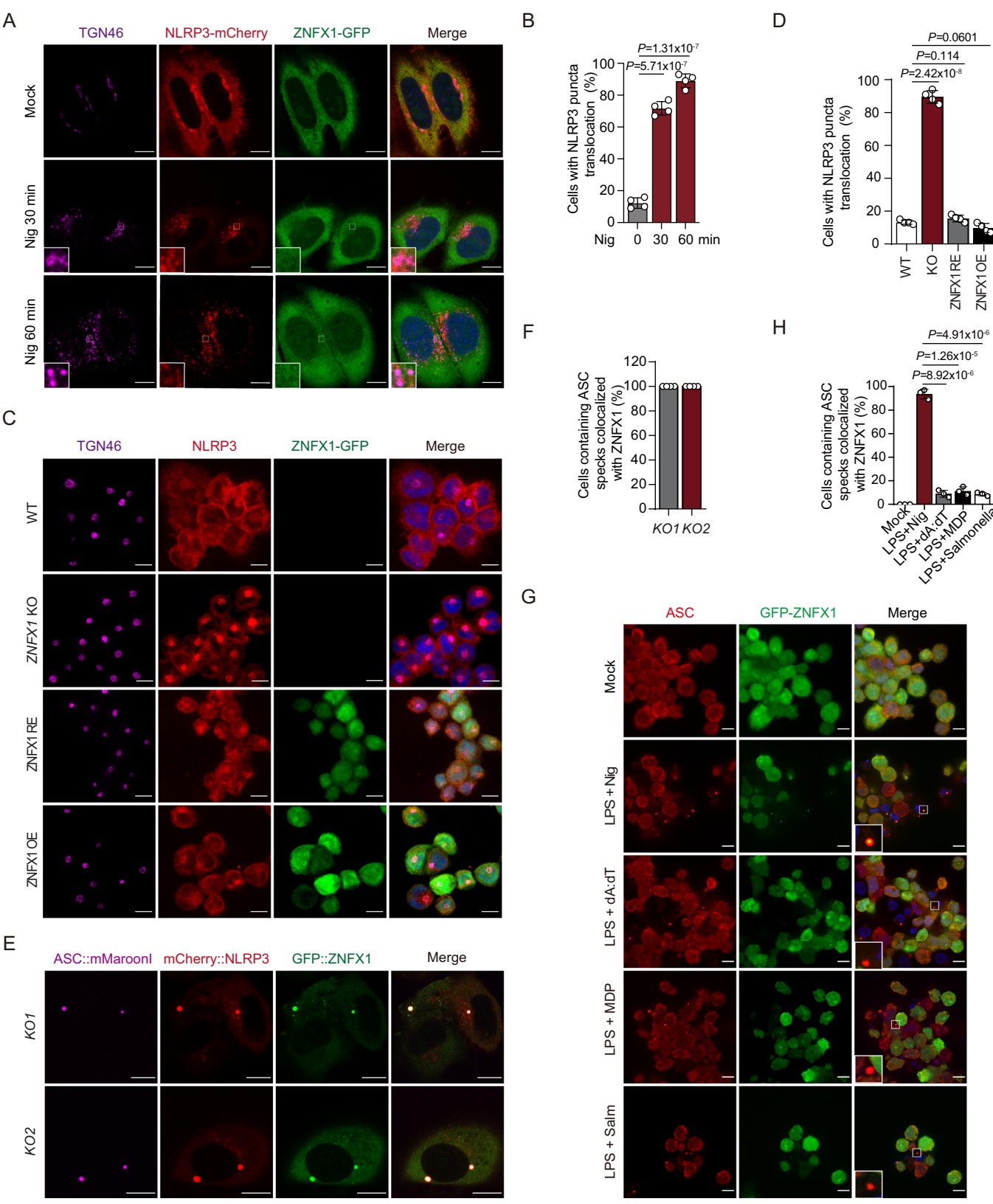

◄ **Figure 4.   ZNFX1 prevents the translocation of NLRP3 to TGN46+ vesicles likely corresponding to TGN.**

(A) HeLa cells stably expressed GFP-ZNFX1 and mCherry-NLRP3 were treated with nigericin for the indicated times, fixed, and stained with anti-TGN46 antibody. Scale bar, 10 μm. (B) Quantification of the percentage of cells showing NLRP3 translocation from 100 cells in (A). $n = 4$ biological replicates. Data were presented as mean ± s.d., two-sided Student's *t*-test. (C) WT, *ZNFX1* KO, *ZNFX1* rescue, and *ZNFX1* OE THP-1-derived macrophages were fixed and stained with anti-TGN46 and anti-NLRP3 antibodies, followed by secondary antibodies conjugated with Alex fluor-647 and Alex fluor-568. Scale bar, 10 μm. (D) Quantification of the percentage of cells showing NLRP3 puncta on TGN46+ vesicles from 100 cells in (C). $n = 4$ biological replicates. Data were presented as mean ± s.d., two-sided Student's *t*-test. (E) *ZNFX1* KO HeLa cells were co-expressed with mMaroon1-ASC, mCherry-NLRP3, and GFP-ZNFX1. Fluorescence imaging was captured under a 63X objective using Leica SP8 laser scanning microscopy. Scale bar, 10 μm. (F) Quantification of cells containing colocalized proteins in (E). $n = 4$ biological replicates. Data were presented as mean ± s.d., two-sided Student's *t*-test. (G) *ZNFX1* rescue THP-1-derived macrophages were treated with the indicated inflammasome agonists (Nig, 45 min; polydA:dT, 12 h; MDP, 16 h; salm, 16 h). Cells were fixed, and GFP-ZNFX-1 and ASC specks were detected with GFP-based fluorescence and anti-ASC-based immunofluorescence, respectively. Inset shows ASC specks and GFP-ZNFX1. Scale bar, 10 μm. (H) Quantification of cells with ASC specks containing GFP-ZNFX1 in (G) from 100 cells. $n = 3$ biological replicates. Data were presented as mean ± s.d., two-sided Student's *t*-test. Source data are available online for this figure.

## ZNFX1 prevents the translocation of NLRP3 to TGN38+/TGN46+ vesicles, likely constituting the TGN

Previous studies have shown that NLRP3 undergoes a series of translocations during inflammasome activation. In the resting state, there is an equilibrium between smaller NLRP3 species and membrane-bound NLRP3 cage-like structures. These cage-like structures are associated with TGN38+/TGN46+ vesicles, which are likely constituents of the TGN (Andreeva et al, 2021). These vesicles continuously shuttle between the cytoplasmic membrane, endosomes, and the TGN. However, NLRP3 agonists disrupt this transport, impeding the retrograde movement of endosomes to the TGN. This disruption leads to the accumulation of NLRP3 on dispersed TGN38+/TGN46+ endosomal vesicles (Andreeva et al, 2021; Chen and Chen, 2018). The translocation of NLRP3 to both the TGN and these dispersed vesicles represents a common early event in NLRP3 inflammasome activation. To investigate the dynamics of NLRP3 translocation, we treated HeLa cells stably expressing GFP-ZNFX1 and mCherry-NLRP3 with nigericin. Time-lapse imaging revealed that mCherry-NLRP3 translocated to dispersed TGN46+ vesicles, while GFP-ZNFX1 predominantly remained diffuse (Fig. 4A,B; Movies EV1–4). Immunofluorescence analysis using an anti-α-TGN46 antibody revealed that NLRP3 translocated to the TGN46+ vesicles  in ~10% of the resting macrophages derived from THP-1 cells (Fig. 4C,D). However, the degree of translocation increased significantly to ~75% in *ZNFX1* KO cells (Fig. 4C,D). Conversely, the percentage of ZNFX1 RE cells exhibiting NLRP3 translocation was reduced to ~10%, and the percentage of ZNFX1 OE cells exhibiting NLRP3 translocation was further reduced to less than 5% (Fig. 4C,D). These results were confirmed using immunofluorescence with an anti-α-TGN38 antibody (Appendix Fig. S3A,B). The observed NLRP3 species likely corresponded to NLRP3 that formed cage-like structures bound to the TGN and were poised for further activation (Andreeva et al, 2021; Chen and Chen, 2018). These findings suggest that ZNFX1 retains NLRP3 in the cytoplasm and prevents its translocation to membrane vesicles in the resting state.

Previous studies have suggested that HeLa cells do not express ASC (Bauernfeind et al, 2009; Fernandes-Alnemri et al, 2009). The observation that NLRP3 translocated to TGN46+/TGN38+ membrane vesicles while ZNFX1 remained diffuse in the cytoplasm in response to nigericin treatment suggests that an event occurred before the formation of ASC specks (Fig. 4A,B; Movies EV1–4). When GFP-ZNFX1, mCherry-NLRP3, and mMaroon1-ASC were co-expressed in HeLa cells, they formed large aggregates resembling ASC specks (Fig. 4E,F; Appendix Fig. S3C,D). Consistent with these findings, ZNFX1 localized to ASC specks specifically when the NLRP3 inflammation, but not the AIM2, NLRP1b, or NLRC4 inflammasome, was activated in THP-1 macrophages (Figs. 1E, 4G,H and EV2C). This suggests that as the ZNFX1 might be recruited to the activated NLRP3 inflammasome, likely exerting a direct inhibitory effect on it. Collectively, these findings indicate that ZNFX1 plays a role in retaining smaller NLRP3 species in the cytoplasm, preventing their translocation to the TGN and the subsequent formation of larger cage-like structures.

## The decrease in ZNFX1 protein levels during inflammation activation is likely mediated by caspase-1 cleavage but not proteasome- or autophagy-mediated protein degradation

Our investigations revealed that the protein level of full-length ZNFX1 in macrophage lysates decreased upon activation of the NLRP3 inflammasome but not upon activation of the AIM2, NLRP1b, or NLRC4 inflammasome (Fig. 1A–D,G–K; Appendix Fig. S1A–E). This decrease in ZNFX1 protein levels could be attributed to (1) decreased solubility of ZNFX1 due to aggregation, leading to the separation of ZNFX1 into pellets during sample preparation; (2) increased protein degradation by either the 26S proteasome pathway or autophagy; and (3) cleavage of ZNFX1 by proteases.

To determine the mechanism underlying the reduction in ZNFX1 protein levels, THP-1-derived macrophages or iBMDMs were stimulated with nigericin or gramicidin, and cell lysates were collected at different time points. Immunoblot analysis of the supernatants and pellets isolated from these lysates showed that NLRP3 agonist treatment led to a time-dependent decrease in the protein levels of ZNFX1, ASC, and NLRP3 in the supernatant, indicating a decrease in the levels of their soluble forms (Figs. 5A,B and EV4A,B). Conversely, the protein levels of ASC and NLRP3 in the cell pellets increased with increasing treatment time, indicating the formation of NLRP3 and ASC oligomers (Figs. 5A,B and EV4A,B). Notably, ZNFX1 was not detected in the cell lysate pellets (Figs. 5A,B and EV4A,B). Interestingly, when THP-1 macrophages were treated with poly(dA:dT) and subjected to the same analysis, elevated ASC protein levels in the pellets and decreased levels in the supernatants were detected, while ZNFX1 and NLRP3 protein levels remained unchanged (Fig. EV4C). This confirmed that the reduction in the level of the full-length ZNFX1 protein specifically occurred during NLRP3 activation. Furthermore, ZNFX1 mRNA and protein levels were increased during the LPS-induced priming of the NLRP3 inflammasome,

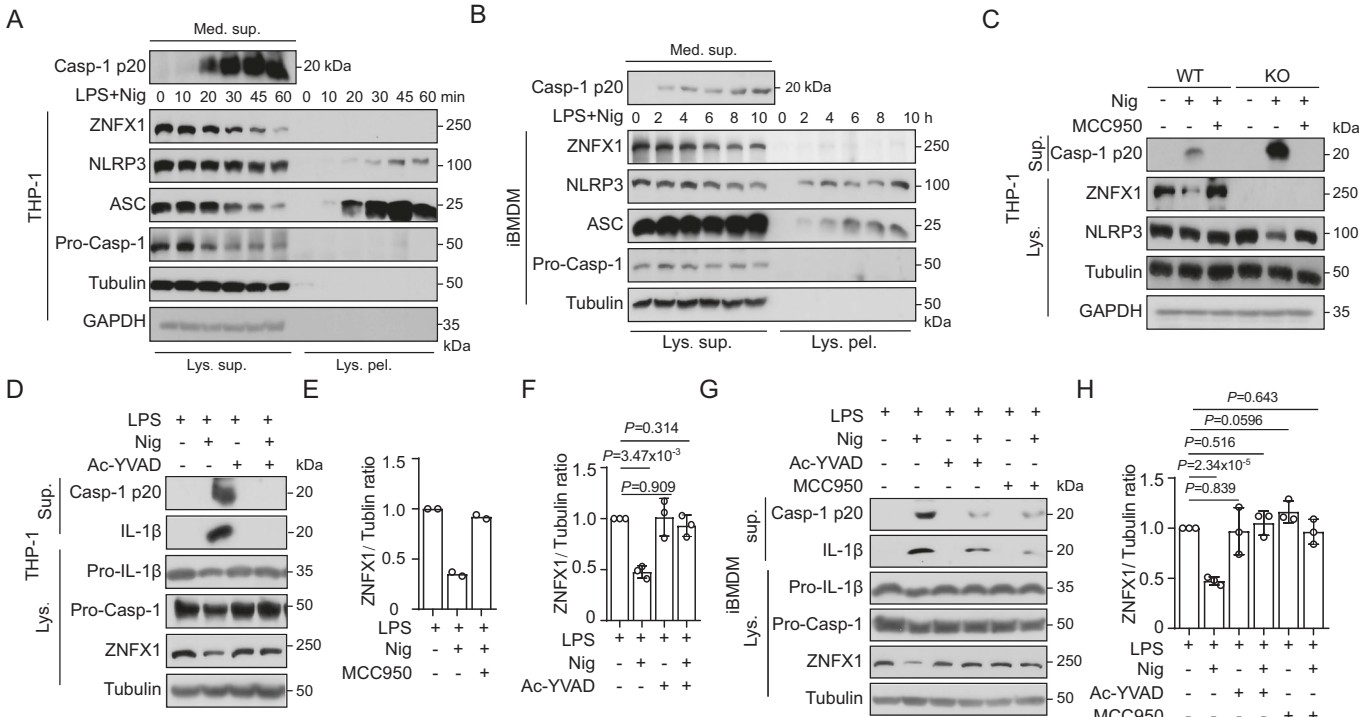

**Figure 5. The decrease of *ZNFX1* protein level during NLRP3 inflammasome activation is likely mediated by caspase-1 cleavage.**

(A, B) LPS-primed THP-1-derived macrophages or iBMDM were treated with nigericin for the indicated time. Cells were lysed, and the supernatant and pellet of the cell lysate were separated by centrifugation as described in the material and methods. Proteins were detected by immunoblot using the indicated antibodies. (C, D) LPS-primed WT or *ZNFX1 KO* THP-1-derived macrophages were treated with nigericin with or without MCC950 or caspase-1 inhibitor (Ac-YVAD-cmk) pretreatment. Proteins from cell lysate or culture medium supernatants were analyzed by immunoblotting. (E, F) Quantification of ZNFX1 levels using ImageJ from (C, D). $n = 2$ biological replicates for (E). $n = 3$ biological replicates for (F). Data were presented as mean ± s.d., two-sided Student's *t*-test. (G) LPS-primed iBMDM cells were treated with nigericin, with or without MCC950 or caspase-1 inhibitor pretreatment. Proteins from cell lysate or culture medium supernatant were detected by immunoblotting. (H) Quantification of ZNFX1 levels using ImageJ from (G). $n = 3$ biological replicates. Data were presented as mean ± s.d., two-sided Student's *t*-test. For each biological replicate in (E, F, H), band intensity was measured using ImageJ. The WT control was set to 1, and the ratio for *ZNFX1* KO cells was calculated by dividing their intensity by the corresponding WT control intensity. Source data are available online for this figure.

consistent with the finding that ZNFX1 functions as an ISG (Fig. EV4D,E). However, the mRNA levels of *ZNFX1* showed minimal changes during the activation of the NLRP3 inflammasome, suggesting that ZNFX1 protein levels were predominantly regulated at the posttranscriptional level (Fig. EV4E,F).

To determine whether ZNFX1 is degraded during NLRP3 inflammasome activation, THP-1-derived macrophages or iBMDMs were pretreated with the proteasome inhibitor MG132 or the macroautophagy inhibitor CQ or 3-MA before NLRP3 inflammasome activation. We found that the protein levels of ZNFX1, NLRP3, IL-1β, and caspase-1 P20 were increased upon MG132, CQ, or 3-MA treatment, as expected and consistent with previous reports (Han et al, 2019; Song et al, 2016). Remarkably, the level of full-length ZNFX1 significantly decreased regardless of the inhibitor that was administered (Fig. EV4G,H). Finally, we discovered that MCC950, a selective NLRP3 inhibitor, and Ac-YVAD-cmk, a specific caspase-1 inhibitor, effectively suppressed the reduction in the level of the full-length ZNFX1 protein during NLRP3 inflammasome activation (Figs. 5C–H and EV4I–K). In summary, these results indicate that the decrease in ZNFX1 expression during NLRP3 inflammasome activation is likely due to caspase-1 cleavage rather than proteasome- or autophagy-mediated protein degradation.

## Cleavage of ZNFX1 by caspase-1 triggers a feed-forward loop that drives NLRP3 inflammasome activation

Our analysis supports the hypothesis that activated caspase-1 cleaves ZNFX1 to relieve its inhibition of NLRP3 inflammasome activation. Three findings support this notion. First, we observed that caspase-1 efficiently cleaved purified ZNFX1 into fragments in vitro (Fig. 6A). Second, upon reestablishment of the NLRP3 inflammasome pathway in HeLa cells, caspase-1-dependent cleavage of ZNFX1 was observed (Fig. EV4L). Third, ZNFX1 fragments corresponding to the cleaved fragments observed in vitro were detected after nigericin-induced NLRP3 activation (Figs. 6B,C and EV4M).

To identify the site of ZNFX1 that is cleaved by caspase-1, we used SitePrediction to predict the potential cleavage sites targeted by caspase-1 around the major cleavage fragments. The 5 potential sites that were identified (D768, D811, D830, E814, and E815) were subsequently mutated to alanine to generate ZNFX1(5D/EA). Interestingly, ZNFX1(5D/EA) was resistant to caspase-1-mediated cleavage in vitro (Fig. 6D). When these sites were individually mutated, only ZNFX1(D830A) was able to resist caspase-1-mediated cleavage (Figs. 6D and EV4N). Thus, D830 is likely a

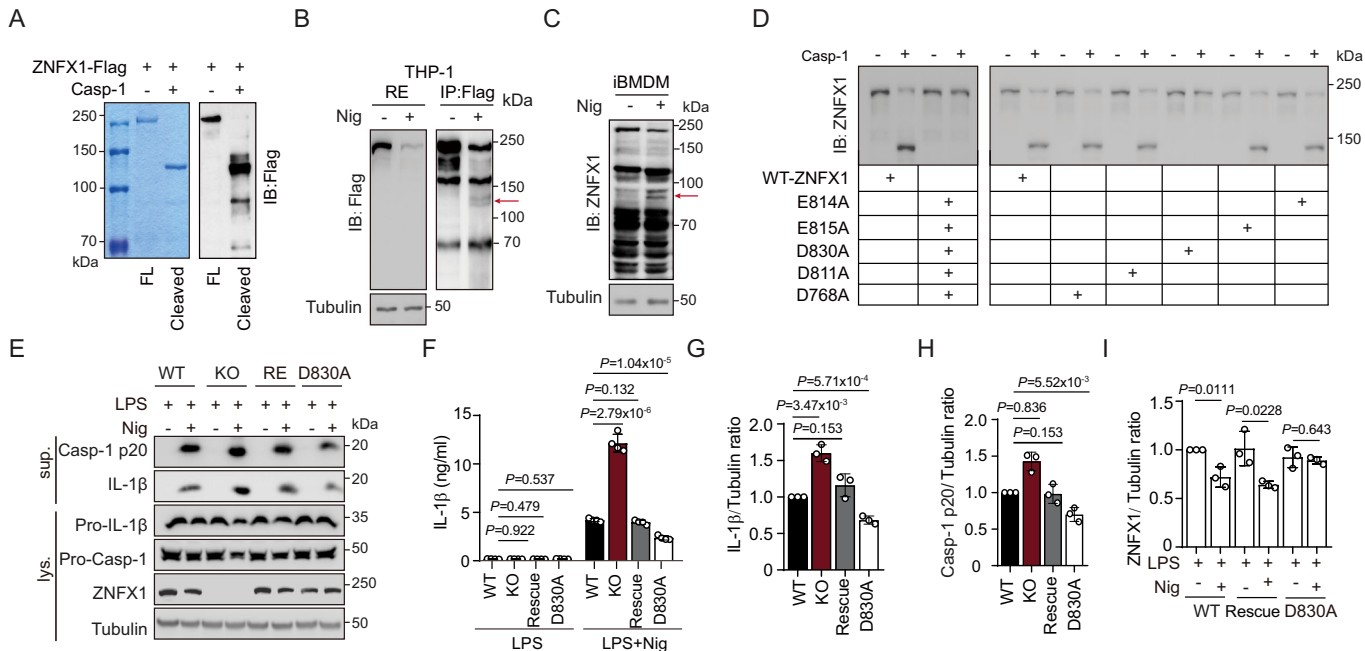

**Figure 6. A feed-forward loop releases ZNFX1's inhibitory role on the NLRP3 inflammasome.**

(A) 3xFLAG-mEGFP-ZNFX1 was purified with α-FLAG beads and incubated with caspase-1 protein. Products were visualized by Coomassie Brilliant Blue staining (left) or immunoblotted with α-FLAG antibody (right). $n = 2$ biological replicates. (B) Cell lysate from LPS + nigericin-treated *ZNFX1* Rescue THP-1 cells were precipitated with α-FLAG antibody. Precipitated ZNFX1 was visualized by immunoblotting. The red arrow indicates putative cleaved ZNFX1 fragments. $n = 2$ biological replicates. (C) Proteins from cell lysate of LPS + nigericin-treated iBMDM cells were detected by immunoblotting. The red arrow indicates putative cleaved ZNFX1 fragments. $n = 2$ biological replicates. (D) Purified 3xFLAG-mEGFP-ZNFX1, with or without potential caspase-1 cleavage sites mutated, was incubated with caspase-1 protein. Products were detected with immunoblotting using α-FLAG antibody. (E–I) WT, *ZNFX1* KO, *ZNFX1* Rescue, and *ZNFX1*(D830A) THP-1 cells-derived macrophages were primed with LPS and activated with nigericin. Proteins from supernatant and cell lysate were visualized using the indicated antibodies (E). IL-1β levels in the supernatant were detected by ELISA (F). The protein level of IL-1β (G) and caspase-1 P20 (H) in the supernatant, and ZNFX1 in cell lysate (I) were quantified using ImageJ. $n = 3$ biological replicates. Data were presented as mean ± s.d., two-sided Student's *t*-test. For each biological replicate in (G–I), band intensity was measured using ImageJ. The WT control was set to 1, and the ratio for *ZNFX1* KO cells was calculated by dividing their intensity by the corresponding WT control intensity. Source data are available online for this figure.

critical cleavage site for caspase-1, although we cannot rule out the possibility that other cleavage sites exist.

We hypothesize that caspase-1-mediated cleavage of ZNFX1 is important for further activation of the NLRP3 inflammasome. To test this hypothesis, we induced the expression of ZNFX1(D830A) in *ZNFX1* KO THP-1 cells. LPS-primed WT, *ZNFX1* KO, ZNFX1 RE, and ZNFX1(D830A) THP-1 cells were exposed to nigericin (Fig. 6E). Immunoblotting and ELISA showed that the levels of IL-1β and caspase-1 P20 were significantly lower in ZNFX1(D830A) cells than in WT and ZNFX1 RE cells (Fig. 6E–H). Additionally, the ZNFX1 protein level decreased in WT and ZNFX1 RE cells but not in ZNFX1(D830A) cells upon nigericin treatment (Fig. 6E,I). Taken together, these results indicate that caspase-1-mediated cleavage of ZNFX1 serves as a trigger for a feed-forward loop that enhances NLRP3 inflammasome activation.

## Pathogenic mutation of ZNFX1 prevents its ability to inhibit the NLRP3 inflammasome

To establish a direct connection between mutations in ZNFX1 found in human patients and excessive inflammation, we generated *ZNFX1* KO THP-1 cell lines expressing either WT ZNFX1 or ZNFX1 with three missense mutations (R334Q, L1051P, and I1154T) found in patients (Vavassori et al, 2021), using lentiviral complementation. Immunoblotting confirmed comparable levels of

ZNFX1 expression in both WT and mutant cells (Fig. 7A,B). Subsequently, THP-1 macrophages were treated with nigericin, and we observed greater caspase-1 cleavage in *ZNFX1* KO cells and cells with mutant ZNFX1 than in WT and ZNFX1 RE cells (Fig. 7B,C). Furthermore, we observed that ZNFX1 containing the patient mutations failed to inhibit the translocation of NLRP3 to TGN38+/TGN46+ vesicles. Approximately 75% of cells harboring these mutations exhibited premature translocation of NLRP3 under resting conditions, whereas only 10% of WT ZNFX1 RE cells exhibited this phenomenon (Fig. 7D,E; Appendix Fig. S4A,B). Immunoprecipitation of mutant ZNFX1 did not result in NLRP3 pulldown, indicating that these mutations disrupted the interaction between ZNFX1 and NLRP3, consequently preventing the sequestration of NLRP3 in the cytoplasm (Fig. 7F).

ZNFX1 is a highly conserved SF1 RNA helicase that binds ATP via its Walker A motif and hydrolyzes ATP through its Walker B motif (Babst et al, 1998; Hanson and Whiteheart, 2005; Sandall et al, 2020). To investigate the role of ZNFX1 helicase activity in inhibiting NLRP3 inflammasome activation, we induced the expression of ZNFX1 containing mutations in either the Walker A motif (K625A) or Walker B motif (E1007Q) into ZNFX1 KO THP-1 cells (Fig. EV5A). We confirmed that these helicase mutants did not affect ZNFX1 expression levels, and we subsequently treated the cells with nigericin. Immunoblotting and

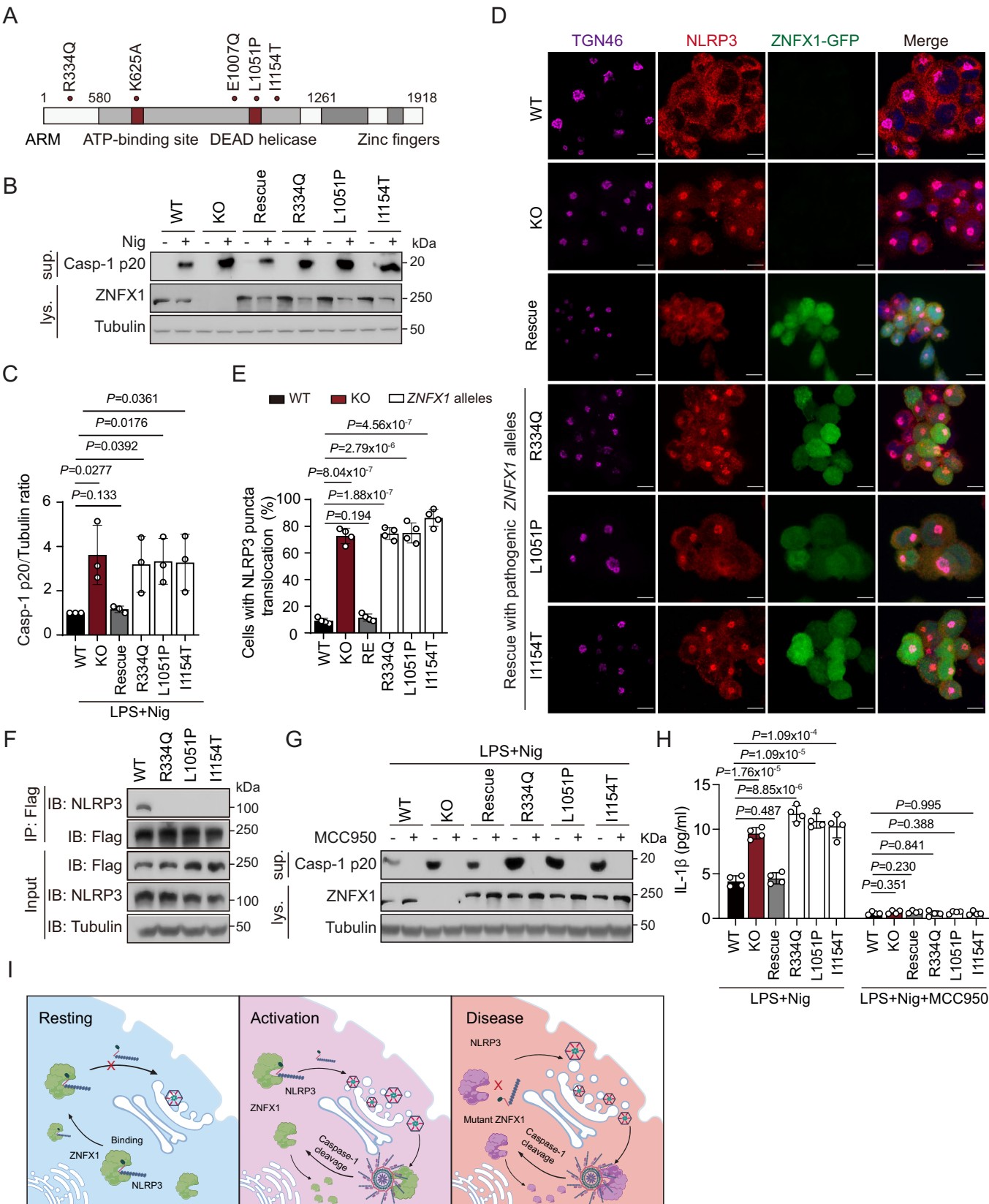

Figure 7.  ZNFX1 proteins with pathogenic lesions fail to inhibit NLRP3 inflammasome.

(A) Predicted domains and identified variants in the ZNFX1 amino acid sequence. (B) The *ZNFX1* open reading frame containing three nonsynonymous patient mutations was reintroduced to *ZNFX1* KO THP-1 cells through lentivirus-mediated delivery. Cells were primed with LPS and stimulated with nigericin for 1 h. Proteins in the culture medium supernatant and cell lysate were detected using the indicated antibodies. (C) Quantification of caspase-1 P20 level in (B). $n = 3$ biological replicates. Data were presented as mean ± s.d., two-sided Student's *t*-test. For each biological replicate, band intensity was measured using ImageJ. The WT control was set to 1, and the ratio for *ZNFX1* KO cells was calculated by dividing their intensity by the corresponding WT control intensity. (D) WT, *ZNFX1* RE, and *ZNFX1* mutants THP-1 derived macrophages were fixed and stained with anti-TGN46 and anti-NLRP3 antibodies, followed by Alex fluor-647 and Alex fluor-568 conjugated secondary antibodies. Scale bar, 10 μm. (E) Quantification of the percentage of cells with NLRP3's translocation to the TGN from 100 cells in (D). $n = 4$ biological replicates, mean ± s.d., two-sided Student's *t*-test. (F) *ZNFX1* KO THP-1 derived macrophages rescued with WT or pathogenic alleles of *ZNFX1* were immunoprecipitated with FLAG-M2 beads. Proteins in input cell lysate and immunoprecipitate were detected using indicated antibodies. (G, H) Cells with the indicated genotype were primed with LPS and treated with nigericin, with or without MCC950 pretreatment. NLRP3 inflammation activation was measured by immunoblot in (G) and ELISA in (H). $n = 4$ biological replicates for (H), mean ± s.d., Student's *t*-test, two-tailed. (I) Working model for *ZNFX1* in inhibiting NLRP3 inflammasome. In the resting stage, ZNFX1 directly interacts with smaller NLRP3 species to sequester NLRP3 in the cytoplasm, preventing its translocation to TGN vesicles to form cage-like structures. NLRP3 agonists disrupt the NLRP3-ZNFX1 interaction, promoting NLRP3 translocation to the TGN and initiating NLRP3 inflammasome activation. After activation, ZNFX1 is likely recruited to the mature NLRP3 inflammasome, where caspase-1 cleaves ZNFX1 to fragments, further releasing NLRP3 and forming a feed-forward loop. ZNFX1 with missense human pathogenic lesions cannot interact with NLRP3, presumably due to conformational changes, leading to hyperinflammation-related diseases when stimulated by NLRP3 activators. Source data are available online for this figure.

immunofluorescence analyses revealed heightened caspase-1 activation and enhanced NLRP3 translocation to the TGN38+/TGN46+ vesicles in cells expressing these helicase mutants compared to WT or ZNFX1 RE cells (Fig. EV5A–D; Appendix Fig. S4A,B). Additionally, the helicase mutants failed to interact with NLRP3 (Fig. EV5E). These findings suggest that the helicase activity of ZNFX1 plays a critical role in suppressing NLRP3 inflammasome activation.

Crucially, treatment with the specific NLRP3 inhibitor MCC950 abolished excess caspase-1 cleavage and IL-1β secretion in THP-1 cells with ZNFX1 mutations found in patients and critical for helicase activity (Figs. 7G,H and EV5F,G). Furthermore, MCC950 treatment reduced the severity of EAE in *Znfx1*$^{-/-}$ mice, as indicated by a decrease in the disease index score and a reduction in IL-1β production in the serum (Appendix Fig. S5A–C). These results support the idea that excess NLRP3 inflammation may be a direct result of patient mutations in ZNFX1, which is consistent with clinical reports. Moreover, our findings suggest that specific inhibitors targeting NLRP3 may have the potential to treat diseases caused by ZNFX1 loss.

## Discussion

In response to pathogenic infections, the intricate interplay between different innate immune response pathways is crucial for defining cell fate and regulating the magnitude of the immune response at various levels, from cellular to systemic (Onomoto et al, 2014; Paget et al, 2023). In this study, we identified the conserved RNA helicase ZNFX1 as a significant regulator that coordinates a balanced innate immune response to determine cell fate.

Our previous research identified ZNFX1 as a dsRNA sensor and an ISG that plays a vital role in combating RNA virus infections by amplifying antiviral interferon responses (Wang et al, 2019). The present study expands the role of ZNFX1 to include the regulation of infections caused by bacterial or fungal pathogens beyond viral infections. As an ISG, ZNFX1 is rapidly upregulated together with NLRP3 during the early stages of infection, keeping NLRP3 in check. This observation highlights how the innate immune system efficiently suppresses the inflammatory response in the early phase of infection. Additionally, we have discovered a novel role for

ZNFX1 in regulating inflammasome activation. Our results suggest that ZNFX1 acts as a cytoplasmic sequester of NLRP3 by interacting with its LRR and NACHT domains. Moreover, ZNFX1 is likely recruited to the active NLRP3 inflammasome. This recruitment event may serve two purposes: First, ZNFX1 can hinder the NLRP3 inflammasome by directly interacting with its active form. Second, ZNFX1 can be cleaved by caspase-1 in response to NLRP3 agonists, but not AIM2, NLRP1b, or NLRC4 agonists, thus indicating a specific cleavage mechanism (Figs. 4G,H and 7I). Together, these findings enhance our understanding of inflammasome regulation by interferons and complement previous reports demonstrating that type I interferon can inhibit NLRP3 inflammasome activation by downregulating pro-IL-1β and pro-IL-18 (Guarda et al, 2011).

Gain-of-function mutations in NLRP3 lead to abnormal spontaneous activation of the NLRP3 inflammasome, causing the dominant systemic autoinflammatory disease known as cryopyrin-associated periodic syndrome (CAPS) (Booshehri and Hoffman, 2019). Conversely, ZNFX1 deficiency is associated with familial immunodeficiency and multisystem inflammatory disease (Vavassori et al, 2021). However, spontaneous inflammasome activation was not observed in either *ZNFX1* KO cell lines or *Znfx1* KO mice during our study. This discrepancy may be attributed to the common early inhibition of NLRP3 translocation by ZNFX1, the loss of which is required but not sufficient for NLRP3 inflammasome activation. ZNFX1 deficiency results in the premature translocation of NLRP3 to TGN38+/TGN46+ vesicles constituting the TGN, which may correspond to NLRP3 species that form cage-like structures, which occurs early in NLRP3 inflammasome activation (Andreeva et al, 2021). Moreover, other studies have indicated that recruiting NLRP3 to the intact TGN is sufficient for subsequent inflammasome formation independent of an additional priming stimulus (Schmacke et al, 2022). Either NLRP3 activators act after NLRP3 translocation but before ASC speck formation, or other negative regulators of NLRP3 need to be removed for NLRP3 inflammasome activation (Xu and Nunez, 2023). Therefore, the loss of ZNFX1 renders patients susceptible to hyperinflammation upon exposure to NLRP3 agonists.

There is a limited range of clinical interventions available for the treatment of familial immune-related diseases resulting from ZNFX1 deficiency. Immunosuppressants, including ruxolitinib, a

JAK inhibitor, were administered to patients but were found to have only a transient ability to relieve symptoms (Vavassori et al, 2021). An alternative therapeutic approach for treating diseases associated with ZNFX1 deficiency could involve pharmacological suppression of the NLRP3 inflammasome, such as through the use of anti-IL-1 therapy in CAPS patients (Booshehri and Hoffman, 2019). In summary, our study offers novel insights into the fundamental aspects of innate immunity with potential relevance for inherited human diseases.

# Methods

### Reagents and tools table

| Reagent/resource | Reference or source | Identifier or catalog number |
|---|---|---|
| **Experimental Models** | | |
| HeLa (*H. sapiens*) | ATCC | CRM-CCL-2 |
| HEK293T (*H. sapiens*) | ATCC | CRL-3216 |
| THP-1 (*H. sapiens*) | ATCC | TIB-202 |
| iBMDM (*M. musculus*) | Feng Shao Laboratory (National Institute of Biological Sciences, Beijing, China) | N/A |
| BMDM (*M. musculus*) | C57BL/6 mice | N/A |
| C57BL/6 *Znfx1*⁻/⁻ mice (*M. musculus*) | Wang et al, 2019 | N/A |
| C57BL/6 J mice (*M. musculus*) | Wang et al, 2019 | N/A |
| **Recombinant DNA** | | |
| lentiCRISPRv2 | Addgene | 52961 |
| psPAX2 | Addgene | 12260 |
| pMD2.G | Addgene | 12259 |
| pcDNA3.1-3XFLAG-mEGFP | This study | N/A |
| pcDNA3.1-3XFLAG-mEGFP-ZNFX1 | This study | N/A |
| pcDNA3.1-3XFLAG-mEGFP-ZNFX1- Walker A mutant | This study | N/A |
| pcDNA3.1-3XFLAG-mEGFP-ZNFX1- Walker B mutant | This study | N/A |
| pcDNA3.1-3XFLAG-mEGFP-NLRP3 | This study | N/A |
| pcDNA3.1-HA-mCherry-NLRP3 | This study | N/A |
| pcDNA3.1-HA-mCherry-ZNFX1 | This study | N/A |
| pcDNA3.1-3XFLAG-mEGFP-CASP-1 | This study | N/A |
| pcDNA3.1-3XFLAG-mEGFP-AIM2 | This study | N/A |

| Reagent/resource | Reference or source | Identifier or catalog number |
|---|---|---|
| pcDNA3.1-3XFLAG-mEGFP-ASC | This study | N/A |
| pcDNA3.1-3XFLAG-mEGFP-NEK7 | This study | N/A |
| pcDNA3.1-HA-mEGFP-NEK7 | This study | N/A |
| pcDNA3.1-3XFLAG-mEGFP-NLRP3-1-93 | This study | N/A |
| pcDNA3.1-3XFLAG-mEGFP-NLRP3-94-1036 | This study | N/A |
| pcDNA3.1-3XFLAG-mEGFP-NLRP3-1-741 | This study | N/A |
| pcDNA3.1-3XFLAG-mEGFP-NLRP3-742-1036 | This study | N/A |
| pcDNA3.1-3XFLAG-mEGFP-NLRP3-94-741 | This study | N/A |
| pcDNA3.1-3XFLAG-mEGFP-NLRP3-220-536 | This study | N/A |
| pcDNA3.1-3XFLAG-ZNFX1-1-580 | Wang et al, 2019 | N/A |
| pcDNA3.1-3XFLAG-ZNFX1-580-1260 | Wang et al, 2019 | N/A |
| pcDNA3.1-3XFLAG-ZNFX1-580-1918 | Wang et al, 2019 | N/A |
| pcDNA3.1-3XFLAG-ZNFX1-1-1260 | Wang et al, 2019 | N/A |
| pcDNA3.1-3XFLAG-ZNFX1-1261-1918 | Wang et al, 2019 | N/A |
| pcDNA3.1-mMaroon1-ASC | This study | N/A |
| pLenti-mMaroon1-TGN46 | This study | N/A |
| pLenti-3xFLAG-mEGFP-ZNFX1 | This study | N/A |
| pLenti-3xFLAG-mEGFP-ZNFX1-R334Q | This study | N/A |
| pLenti-3xFLAG-mEGFP-ZNFX1-L1051P | This study | N/A |
| pLenti-3xFLAG-mEGFP-ZNFX1-I1154T | This study | N/A |
| pLenti-3xFLAG-mEGFP-ZNFX1-mutant walker A | This study | N/A |
| pLenti-3xFLAG-mEGFP-ZNFX1-mutant walker B | This study | N/A |
| pLenti-3xFLAG-mEGFP-ZNFX1-D768A | This study | N/A |

| Reagent/resource | Reference or source | Identifier or catalog number |
| --- | --- | --- |
| pLenti-3xFLAG-mEGFP-ZNFX1-D811A | This study | N/A |
| pLenti-3xFLAG-mEGFP-ZNFX1-D830A | This study | N/A |
| pLenti-3xFLAG-mEGFP-ZNFX1-E814A | This study | N/A |
| pLenti-3xFLAG-mEGFP-ZNFX1-E815A | This study | N/A |
| pLenti-3xFLAG-mEGFP-ZNFX1-5DE/A | This study | N/A |
| pLenti-HA-mCherry-NLRP3 | This study | N/A |
| pLenti-3xFLAG-miniturbo-mEGFP-ZNFX1 | This study | N/A |
| **Antibodies** | | |
| Rabbit anti-ZNFX1 | Dia-An Biotech | N/A |
| Rabbit anti-ZNFX1 | Abcam | ab179452 |
| Mouse anti-NLRP3 | AdipoGen | AG-20B-0014-C100 |
| Rabbit Anti-NLRP3 | Abcam | ab263899 |
| Rabbit Anti-NLRP3 | GeneTex | GTX00763 |
| Mouse anti-Tubulin | Ray antibody | RM2003 |
| Mouse anti-GAPDH | Ray antibody | RM2002 |
| Rabbit anti-caspase-1 | Abcam | 179515 |
| Rabbit anti-NEK7 | Abcam | 133514 |
| Rabbit anti-caspase-1 | Cell Signaling Technology | 2225 |
| Mouse anti-caspase-1 | AdipoGen | AG-20B-0042-C100 |
| Mouse anti-IL-1β | Cell Signaling Technology | 12242 |
| Mouse anti-IL-1β | Santa Cruz | sc-52012 |
| Rabbit anti-ASC | AdipoGen | AG-25B-0006-C100 |
| Rabbit anti-TGN46 | Invitrogen | PA5-23068 |
| Rabbit anti-TGN38 | Novus Biologicals | 03495S |
| Mouse anti-TGN38 | Santa Cruz | sc-166594 |
| Mouse anti-FLAG | Sigma-Aldrich | F1804 |
| Rabbit anti-HA | Cell Signaling Technology | 3724 |
| Mouse IgG | Santa Cruz | sc-2025 |
| Goat anti-rabbit IgG (H/L):HRP | Ray antibody | RM3002 |
| Goat anti-mouse IgG (H/L):HRP | Ray antibody | RM3001 |
| Goat anti-Rabbit IgG (H + L) Cross-Adsorbed Secondary Antibody, Alexa Fluor™ 647 | Invitrogen | A-21244 |
| Goat anti-Mouse IgG (H + L) Cross-Adsorbed Secondary Antibody, Alexa Fluor™ 647 | Invitrogen | A-21235 |
| Goat anti-Rabbit IgG (H + L) Cross-Adsorbed Secondary Antibody, Alexa Fluor™ 568 | Invitrogen | A-11011 |
| Goat anti-Mouse IgG (H + L) Cross-Adsorbed Secondary Antibody, Alexa Fluor™ 568 | Invitrogen | A-11004 |
| Goat anti-Rabbit IgG (H + L) Highly Cross-Adsorbed Secondary Antibody, Alexa Fluor™ 488 | Invitrogen | A-11034 |
| Goat anti-Mouse IgG (H + L) Cross-Adsorbed Secondary Antibody, Alexa Fluor™ 488 | Invitrogen | A-11001 |
| **Oligonucleotides and other sequence-based reagents** | | |
| Human ZNFX1 gRNA1 | This study | 5′-GCGCCAGATCCTACAGAAGG-3′ |
| Human ZNFX1 gRNA2 | This study | 5′-GCCACCAAGAGCTAGAAATC-3′ |
| Mouse *Znfx1* gRNA1 | This study | 5′-GGTACTGGTTCCGAATGTCG-3′ |
| Mouse *Znfx1* gRNA2 | This study | 5′-GTGCTCTCGGTTTGAAACGG-3′ |
| PCR primers | Appendix Table S1 | N/A |
| **Chemicals, enzymes, and other reagents** | | |
| Ac-YVAD-cmk | MedChemExpress | HY-16990 |
| SYBR Green Pro Taq HS Premix | Accurate Biology | AG11701 |
| Fetal bovine serum | ExCell Bio | FSP500 |
| Fetal bovine serum | Gibco | A5670701 |
| Dulbecco's Modified Eagle's Medium | Gibco | C11995500BT |
| Dulbecco's Modified Eagle's Medium | Corning | 10-013-CVRC |
| RPMI 1640 | Gibco | C11875500BT |
| RPMI 1640 | Corning | 10-040-CVRC |
| Trypsin | Gibco | 25200072 |
| PBS, pH 7.4 | Gibco | 10010023 |
| Opti-MEM | Gibco | 11058021 |

| Reagent/resource | Reference or source | Identifier or catalog number |
|---|---|---|
| Penicillin-Streptomycin | Gibco | 15140122 |
| MCC950 | InvivoGen | inh-mcc |
| PMA | Beyotime | S1819 |
| PMA | Sigma-Aldrich | P1585 |
| LPS | Sigma-Aldrich | L7011 |
| LPS | InvivoGen | Ultra-Pure |
| Pam3CSK4 | InvivoGen | tlrl-pms |
| Nigericin | InvivoGen | tlrl-nig |
| Gramicidin | MedChemExpress | HY-P0163 |
| Salmonella | InvivoGen | tlrl-hkst2 |
| Silica | Sigma-Aldrich | 288586 |
| Alum | Thermo Scientific | 77161 |
| MSU | InvivoGen | tlrl-msu |
| MDP | InvivoGen | tlrl-mdp |
| poly(dA:dT) | InvivoGen | tlrl-patn-1 |
| poly(I:C) | InvivoGen | tlrl-pic |
| Lipofectamine 2000 | Invitrogen | 11668019 |
| Lipofectamine 3000 | Invitrogen | L3000015 |
| Mounting medium | Vectashield | H-1000-10 |
| DSS (disuccinimidyl suberate) | Thermo Fisher Scientific | 21655 |
| FLAG-M2 agarose beads | Sigma-Aldrich | A2220 |
| Protein A/G beads | Santa Cruz | sc-2003 |
| FLAG peptide | Sigma-Aldrich | F4799 |
| Streptavidin Magnetic Beads | NEB | S1420S |
| Recombinant human active caspase-1 | Abcam | ab39901 |
| Phenylmethyl sulfonyl fluoride | Sangon Biotech | A100754-0005 |
| 5×Loading buffer(DTT) | Fude Biological Technology | FD006 |
| Triton X-100 | Sigma-Aldrich | T8787 |
| Tween-20 | Beyotime Biotechnology | ST825 |
| Bovine serum albumin | MP Biomedicals | 02FC007780 |
| Paraformaldehyde | Sigma-Aldrich | 16005-250G-F |
| Polybrene | Sigma-Aldrich | TR-1003 |
| Puromycin | Sigma-Aldrich | P9620 |
| cOmplete(TM), EDTA-free Protease Inhibitor Cocktail | Roche | 4693132001 |
| PVDF membrane | Merck millipore | IPVH00010 |
| Biotin | Sigma-Aldrich | 14400 |
| Oligodendrocyte glycoprotein (MOG) peptide | ChinaPeptides | 4010006243 |

| Reagent/resource | Reference or source | Identifier or catalog number |
|---|---|---|
| Mycobacterium tuberculosis | Chondrex | 7001 |
| Pertussis toxin | List Biological Laboratories | 180 |
| Ionomycin | Beyotime | S1672 |
| Brefeldin A | Beyotime | S1536 |
| MG132 | Sigma-Aldrich | SML1135 |
| Chloroquine | Sigma-Aldrich | C6628 |
| 3-Methyladenine | Sigma-Aldrich | M9281 |
| Human IL-1β ELISA kit | BD Biosciences | 557953 |
| Human TNF-α kit | BD Biosciences | 555212 |
| Mouse IL-1β ELISA kit | BD Biosciences | 59603 |
| RevertAid First Strand cDNA Synthesis | Thermo Scientific | K1621 |
| **Software** | | |
| ImageLab (Version 5.2) | https://www.bio-rad.com/ | N/A |
| GraphPad Prism (Version 9.1.0) | https://www.graphpad.com/ | N/A |
| Fiji | https://imagej.net/ | N/A |

## Methods and protocols

### Cell culture

The THP-1, HeLa, and HEK293T cell lines were obtained from ATCC. iBMDM cells were generously donated by the Feng Shao Laboratory (National Institute of Biological Sciences, Beijing, China). THP-1 cells were cultured in RPMI 1640 medium supplemented with 10% FBS. iBMDMs, HeLa, and HEK293T cells were cultured in DMEM medium supplemented with 10% FBS and antibiotics (Penicillin/Streptomycin). All cells were incubated at 37 °C with 5% $CO_2$ in a humidified incubator.

## Inflammasome stimulation

THP-1 cells were differentiated into macrophages by exposure to 100 ng/ml phorbol-12-myristate-13-acetate (PMA) for 12 h, and then the macrophages had a resting period of 24 h before stimulated. Differentiated THP-1 cells were primed with 200 ng/ml LPS (InvivoGen) or 500 ng/ml Pam3CSK4 (for non-canonical inflammasome activation) for 4 h in RPMI 1640 medium, culture medium was then replaced with serum-free Opti-MEM medium containing various inflammasome stimuli with the following concentration and time in the brackets unless otherwise indicated: nigericin (10 μM for 1 h), gramicidin (1 μM for 2 h), Salmonella (1:1000, 16 h), silica (500 μg/ml, 6 h), Alum (250 μg/ml for 6 h), MSU (200 μg/ml for 6 h), MDP (500 ng/ml for 16 h). For poly(dA:dT) or poly(I:C) treatment, cells were transfected with poly (dA:dT) (2 μg/ml) or poly (I:C) (3 μg/ml) using Lipofectamine 2000. iBMDM cells were primed with 1 μg/ml LPS in DMEM medium for 4 h before replaced with Opti-MEM medium supplied with the following inflammation agonists: nigericin (20 μM for 4 h), gramicidin (5 μM for 4 h), silica (1000 μg/ml for 6 h), Alum (500 μg/ml for 6 h), MDP (5 μg/ml

for 16 h), poly (dA:dT) (2 µg/ml for 16 h). After stimulation, culture medium supernatants and cell lysates were collected separately, precipitated supernatants and cell extracts were separated by SDS-PAGE and immunoblotted with anti-ZNFX1(Dia-An Biotech), anti-ZNFX1(Abcam), anti-NLRP3 (AdipoGen), anti-NLRP3 (Abcam), anti-NLRP3 (GeneTex), anti-Tubulin, anti-GAPDH, anti-caspase-1 (Abcam), anti-NEK7, anti-caspase-1 (CST), anti-caspase-1 (AdipoGen), anti-IL-1β (CST), anti-IL-1β (Santa Cruz) antibodies.

## Generating knockout cell lines with CRISPR/Cas9

sgRNAs targeting human or mouse ZNFX1 were cloned into lentiCRISPRv2 to generate lentiCRISPRv2-sgRNAs. lenti-CRISPRv2 or lentiCRISPRv2-sgRNAs, psPAX2, and pMD2.G plasmids were transfected into HEK293T cells using lipofectamine 3000, medium containing lentivirus were harvested 60 h post-transfection, spun down at 3000 rpm for 10 min and filtered with 0.45-µm filter to clear cell debris, lentivirus was stored at −80 °C. HeLa, THP-1, or iBMDM cells were transduced with lentivirus at low MOI supplied with 10 µg/ml polybrene. Puromycin was added to the medium 72 h later, single puromycin resistance cells were sorted by Flow Cytometry. ZNFX1 sequence and expression were then verified in these cell colonies using Sanger sequencing and Western blot.

## Generating stable expression cell lines using lentivirus-mediated transduction

Lentiviruses expressing 3× FLAG-mEGFP-ZNFX1 or 3× FLAG-mEGFP-ZNFX1 harboring nonsynonymous mutations (L1051P, R334Q, I1154T, walker A motif K625A, and walker B motif E1007Q) were generated as described above. ZNFX1 KO THP-1 or iBMDM cell line was transduced with these lentiviruses at low MOI. Single cells stably expressing GFP were sorted with Flow Cytometry. WT and mutant ZNFX1 expression were verified in these colonies by Western Blot. HeLa cells stably expressing GFP-ZNFX1 and mCherry-NLRP3 were generated by transducing ZNFX1 KO HeLa cells with lentivirus expressing GFP-ZNFX1 and mCherry-NLRP3 using a similar approach.

## Immunofluorescence (IF)

Cells growing on a coverslip were fixed with 4% paraformaldehyde in PBS for 20 min at room temperature and washed with PBS three times. Fixed cells were permeabilized with 0.5% Triton X-100 in PBS for 10 min at 4 °C and blocked with 5% goat serum in IF buffer (PBS, 0.1% bovine serum albumin, 0.2% Triton X-100, 0.05% Tween-20) for 1 h at room temperature. Cells were incubated with the following primary antibodies overnight at 4 °C: anti-NLRP3 (AdipoGen, 1:250), anti-ASC, anti-TGN46, anti-TGN38 (Novus Biologicals, 1:100), and anti-ZNFX1(Abcam, 1:250). Cells were washed with IF buffer three times, 10 min per time, and then incubated with secondary antibodies conjugated with Alex fluor 488 or 568 or 647 (1: 500) for 1 h at RT. Cells were washed with IF buffer three times (10 min per time), and mounted with antifade mounting medium.

## ASC oligomerization analysis

THP-1-derived macrophages or iBMDMs were plated into six-well tissue culture plates (~$2 \times 10^6$ cells per well) and stimulated as indicated. After stimulation, cells were lysed in TBS buffer (50 mM Tris-HCl, 150 mM NaCl, pH 7.4) containing 0.5% Triton X-100 on a rocker for 30 min on ice, and then centrifuged at $6000 \times g$ at 4 °C for 15 min. The Triton X-100 soluble supernatants were discarded, and the Triton X-100 insoluble pellets were washed twice with TBS buffer and resuspended in 200 µl TBS buffer. The pellets were then cross-linked at 37 °C for 30 min by adding 2 mM fresh prepared disuccinimidyl suberate (DSS). The cross-linked pellets were spun down at $6000 \times g$ for 15 min, dissolved in 2× SDS loading buffer, and analyzed with Western blot.

## Cytokine measurements

Cytokine production was measured with a human IL-1β ELISA kit, TNF-α kit, and mouse IL-1β ELISA kit according to the manufacturer's instructions.

## SDD-AGE analysis of oligomerization of NLRP3

The oligomerization of NLRP3 was performed as previously described with minor modifications (Hou et al, 2011; Jiang et al, 2017). Cells were lysed with TBS buffer (50 mM Tris-HCl, 150 mM NaCl, PH 7.4) supplied with 0.5% Triton X-100, 1 mM PMSF, and 1 × Roche protease inhibitor cocktail, on a rocker for 30 min on ice. Lysates were cleared by centrifuging at $6000 \times g$ for 15 min at 4 °C, the pellets were resuspended in 1 × Sample buffer (0.5 × TBE, 10% glycerol, 2% SDS, and 0.0025% bromophenol blue). Proteins were then loaded onto a vertical 1% agarose gel and separated by electrophoresis for 1 h at 4 °C in the running buffer (1 × TBE and 0.1% SDS). Agarose gel was transferred to PVDF membrane (Millipore), immunoblotted with anti-NLRP3 antibody.

## Immunoprecipitation

Plasmids encoding 3× FLAG-tagged and HA-tagged proteins were transfected into HEK293T cells with Lipofectamine 3000, and cells were lysed with lysis buffer (50 mM Tris-HCl pH 7.4, 150 mM NaCl, 0.5% Triton X-100) on ice for 30 min. Lysates were cleared by spinning down at $12,000 \times g$ for 10 min. 5% supernatants were saved as input. The rest supernatants were incubated with FLAG-M2 agarose beads for 4 h at 4 °C. The beads were washed with lysis buffer four times and then resuspended in 30 µl 2 × SDS loading buffer. Input and immunoprecipitated proteins were resolved with SDS-PAGE and detected with anti-FLAG, anti-HA, and anti-Tubulin antibodies (DSHB).

For Immunoprecipitation from endogenous proteins, THP-1-derived macrophages or iBMDM cells from two 10 cm culture dishes were lysed with lysis buffer on ice for 30 min. Supernatants were collected by centrifuging and incubated with 4 µl anti-NLRP3 (AdipoGen) or IgG control for 4 h at 4 °C. About 20 µl Protein A/G beads were then added to supernatants and incubated overnight at 4 °C. The beads were washed with lysis buffer six times. Input and immunoprecipitated proteins were analyzed with Western blot.

## Surface plasmon resonance (SPR) analysis

To purify ZNFX1 and NLRP3 proteins, HEK293T cells were transfected with plasmids encoding 3 × FLAG::ZNFX1 or 3 × FLAG::NLRP3 with Lipofectamine 3000. Twenty-four hours later, cells were lysed with lysis buffer on ice for 30 min. Lysates were cleared by spinning down at 12,000 ×$g$ for 10 min at 4 °C. The supernatants were incubated with FLAG-M2 agarose beads for 4 h at 4 °C. The beads were washed with lysis buffer five times. 3 × FLAG::ZNFX1 or 3 × FLAG::NLRP3 were eluted with 150 ng/µl FLAG peptide and concentrated with Amicon Ultra-10 kDa centrifugal filter unit (Merck Millipore, Cat # ACS501024). SDS-PAGE and coomassie blue staining confirmed the purity of 3 × FLAG::ZNFX1 or 3 × FLAG::NLRP3. Purified 3 × FLAG::ZNFX1 or 3 × FLAG::NLRP3 were immobilized on a CM5 chip (GE Healthcare, Cat # 29-1049-88) by using an amino coupling kit (GE Healthcare, Cat # BR-1000-50) following the vendor's instructions. Different concentrations of 3 × FLAG::NLRP3 or 3 × FLAG::ZNFX1 were then added to the chip and analyzed on a Biacore T200 system.

## Proximity biotin labeling

Lentiviruses expressing miniTurbo-mEGFP-ZNFX1 were generated as described above. *ZNFX1* KO THP-1 cell line and *ZNFX1* KO HeLa cells stably expressing mCherry-NLRP3 were transduced with these lentiviruses at low MOI. Cells colonies stably expressing GFP were sorted with Flow Cytometry. miniTurbo-ZNFX1 expression was verified in these colonies by Western Blot. THP-1 or HeLa cells stably expressing miniTurbo-ZNFX1 were plated into 10 cm culture plates and treated with 200 µM biotin for 8 h. Cells were lysed in RIPA buffer by rotating at 4 °C for 1 h. Lysates were spun down at 14,000 × $g$ at 4 °C for 15 min. About 30 µL Streptavidin Magnetic Beads were then added to supernatants and incubated 2 h at 4 °C. The beads were washed with lysis buffer six times. Input and immunoprecipitated proteins were analyzed with Western blot. The beads were washed with RIPA (1 mL, 2 min) ×1 ; 2 M Urea in 10 mM Tris-HCl (pH = 8.0) (1 mL, ~30 s) ×1 ; 2% SDS (1 mL, 2 min) ×2 ; TBST (500 µL, 2 min) ×2 ; TBS (500 µL, 2 min) ×3. 5 mM biotin was used for competitive elution. Input and immunoprecipitated proteins were analyzed with Western blot.

## Protein fractionation by size-exclusion column

Cell extracts prepared from THP-1-derived macrophages or iBMDM cells with RIPA lysis buffer were centrifuged at 20,000 rpm for 10 min at 4 °C. Proteins were concentrated with Amicon® Ultra-30kDa in TBS buffer (50 mM Tris-HCl pH 7.4, 150 mM NaCl). About 7 mg of protein in a volume of 0.5 mL was loaded onto a size-exclusion column (Superdex 200 Increase 10/300 GL) to separate the large protein complexes. Samples were fractionated with a flow rate of 0.25 mL per minute, and elution was collected as 0.5 mL fractions. Protein fractions were separated by SDS-PAGE and detected by western blotting with antibodies against ZNFX1 or NLRP3. Note, the elution of protein fractions of ZNFX1 or NLRP3 from the same experiment were transferred together in one PVDF membrane for comparison.

## Analysis of proteins in supernatants and pellets of cell lysates

THP-1-derived macrophages or iBMDM cells were plated into 6 cm culture plates (~5 ×$10^6$ cells per well), primed with LPS, and treated with NLRP3 inflammasome stimulators for the indicated time. After stimulation, cells were lysed in RIPA buffer (50 mM Tris-HCl pH 8.0, 150 mM NaCl, 1% Triton X-100, 0.1% SDS, 0.5% Sodium deoxycholate) by rotating at 4 °C for 1 h. Lysates were spun down at 14,000 × $g$ at 4 °C for 15 min. The supernatant was transferred to a new tube. The insoluble pellets were washed with RIPA buffer 6 times. The pellets were resuspended in 100 µl 2 × SDS loading buffer and boiled at 100 °C for 10 min. Supernatant and pellets were analyzed with Western blot.

## In vitro assays of ZNFX1 cleavage by caspase

To examine the cleavage of ZNFX1 in vitro, 3.2 µM purified 3 × FLAG-ZNFX1 proteins were incubated with 3U recombinant human active caspase-1 in 25 µL reaction mix containing 50 mM HEPES (pH 7.5), 150 mM NaCl, 3 mM EDTA, 0.005% (v/v) Tween-20, and 10 mM DTT. The reaction was carried out at 37 °C for 6 h. SDS loading buffer was then added to the reaction mixture followed by boiling at 95 °C for 5 min. The samples were analyzed by SDS-PAGE and Western blot.

## qRT-PCR

Total RNA was extracted using the RNAiso Plus reagent, and cDNA was synthesized using the RevertAid First Strand cDNA Synthesis Kit according to the manufacturer's instructions. mRNA levels were quantified by PCR with the SYBR Green Pro Taq HS kit. The primer sequences for mRNA quantification are listed in Appendix Table S1.

## In vivo LPS challenge

Our study examined male and female animals, and similar findings are reported for both sexes. C57BL/6 *Znfx1*[−/−] mice were generated by a CRISPR/Cas9 system from GemParmatech Co as described previously (Wang et al, 2019). Eight to ten weeks male and female C57BL/6J mice were injected intraperitoneally with MCC950 (50 mg/ kg) or vehicle control (DMSO/PBS) 2 h before intraperitoneal injection of LPS (20 mg/kg) (Sigma-Aldrich) or sterile PBS. After 3 h, mice were euthanized, and serum levels of IL-1β and TNF-α were measured by ELISA kits according to the manufacturer's instructions.

## In vivo induction of EAE

Induction and assessment of EAE were performed as previously described with minor modifications (Coll et al, 2015; Shi et al, 2016). Our study exclusively examined female mice because the disease modeled is only relevant in females. Each hind flank of 8–10 weeks female C57BL/6J mice were injected subcutaneously with 250 µg of myelin oligodendrocyte glycoprotein (MOG) peptide 35 to 55 emulsified in complete Freund's adjuvant containing (0.4 mg/mouse) heat-killed Mycobacterium tuberculosis (4 mg/ml) at two different sites, followed by an intraperitoneal injection of

500 ng of pertussis toxin. On day 2, mice were given a second injection of 500 ng of pertussis toxin. Mice were assigned randomly to two groups and respectively administered with THL (2.5 mg/kg) or vehicle control (DMSO/ PBS) via intraperitoneal every second day starting at the induction of the disease on day 0. Mice were weighed every day to assess disease activity, and disease severity was scored as follows: no clinical signs, 0; partially limp tail, 0.5; paralyzed tail, 1; loss in coordinated movement and hind limb paresis, 2; one hind limb paralyzed, 2.5; both hind limbs paralyzed, 3; hind limbs paralyzed and weakness in forelimbs, 3.5; forelimbs paralyzed, 4; and moribund, 5. On day 19 after immunization, mice brain mononuclear cells (MNCs) were isolated using a 30/70% Percoll gradient. MNCs were stimulated for 4 h with PMA (10 ng/ml, Beyotime) and ionomycin (1 µg/ml) in the presence of brefeldin A (5 µg/ml). Cells were stained and analyzed by flow cytometry.

## Ethics approval

The animal experimental procedures and animal care were approved by the Animal Care Committee of the Beijing University of Chinese Medicine, Beijing, China (Approval No.: BUCM–2023112002–4086).

## Statistics and reproducibility

*P* value was calculated with Student's *t*-test (paired, two-tailed) or log-rank test. Each dot represents an individual experiment. The specific tests and numbers of experiments are indicated in the Figure legends.

## Declaration of AI-assisted technologies in the writing process

During the preparation of this manuscript, the authors used ChatGPT 3.5 to polish the text, correct grammar errors, and improve the readability of the manuscript. After using this tool, the authors reviewed and edited the content as needed, and they took full responsibility for the content of the publication.

## Data availability

Our study includes no data deposited in public repositories.

The source data of this paper are collected in the following database record: biostudies:S-SCDT-10_1038-S44318-024-00236-9.

## Peer review information

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

## Acknowledgements

We thank members of the Wan laboratory and Xu laboratory for helpful discussions. This work was supported by the Joint Funds of the National Natural Science Foundation of China (U23A6012) to Anlong Xu, the National Natural Science Foundation of China grants (32070798 and 32370729), the Natural Science Foundation of Guangdong (2024A1515012650), and Shenzhen Medical Research Fund (SMRF No. B2302029) to Gang Wan, the Beijing Nova Program (20230484342), the Young Elite Scientists Sponsorship Program by China Association of Chinese Medicine (2023-QNRC2-A02), the National Natural Science Foundation of China (31900661) to Yao Wang and by Guangdong Science and Technology Department (2023B1212060028).

## Author contributions

**Jing Huang**: Conceptualization; Data curation; Formal analysis; Validation; Investigation; Visualization; Methodology; Writing—review and editing. **Yao Wang**: Conceptualization; Resources; Data curation; Formal analysis; Funding acquisition; Validation; Investigation; Visualization; Methodology; Writing—review and editing. **Xin Jia**: Data curation; Investigation. **Changfeng Zhao**: Data curation; Investigation. **Meiqi Zhang**: Data curation; Investigation. **Mi Bao**: Data curation; Investigation. **Pan Fu**: Data curation; Investigation.

**Cuiqin Cheng**: Data curation; Investigation. **Ruona Shi**: Investigation. **Xiaofei Zhang**: Investigation. **Jun Cui**: Conceptualization; Investigation. **Gang Wan**: Conceptualization; Resources; Data curation; Formal analysis; Supervision; Funding acquisition; Validation; Investigation; Visualization; Methodology; Writing—original draft; Project administration; Writing—review and editing. **Anlong Xu**: Conceptualization; Resources; Data curation; Formal analysis; Supervision; Funding acquisition; Validation; Investigation; Methodology; Project administration; Writing—review and editing.

Source data underlying figure panels in this paper may have individual authorship assigned. Where available, figure panel/source data authorship is listed in the following database record: biostudies:S-SCDT-10_1038-S44318-024-00236-9.

## Disclosure and competing interests statement

The authors declare no competing interests.

# Expanded View Figures

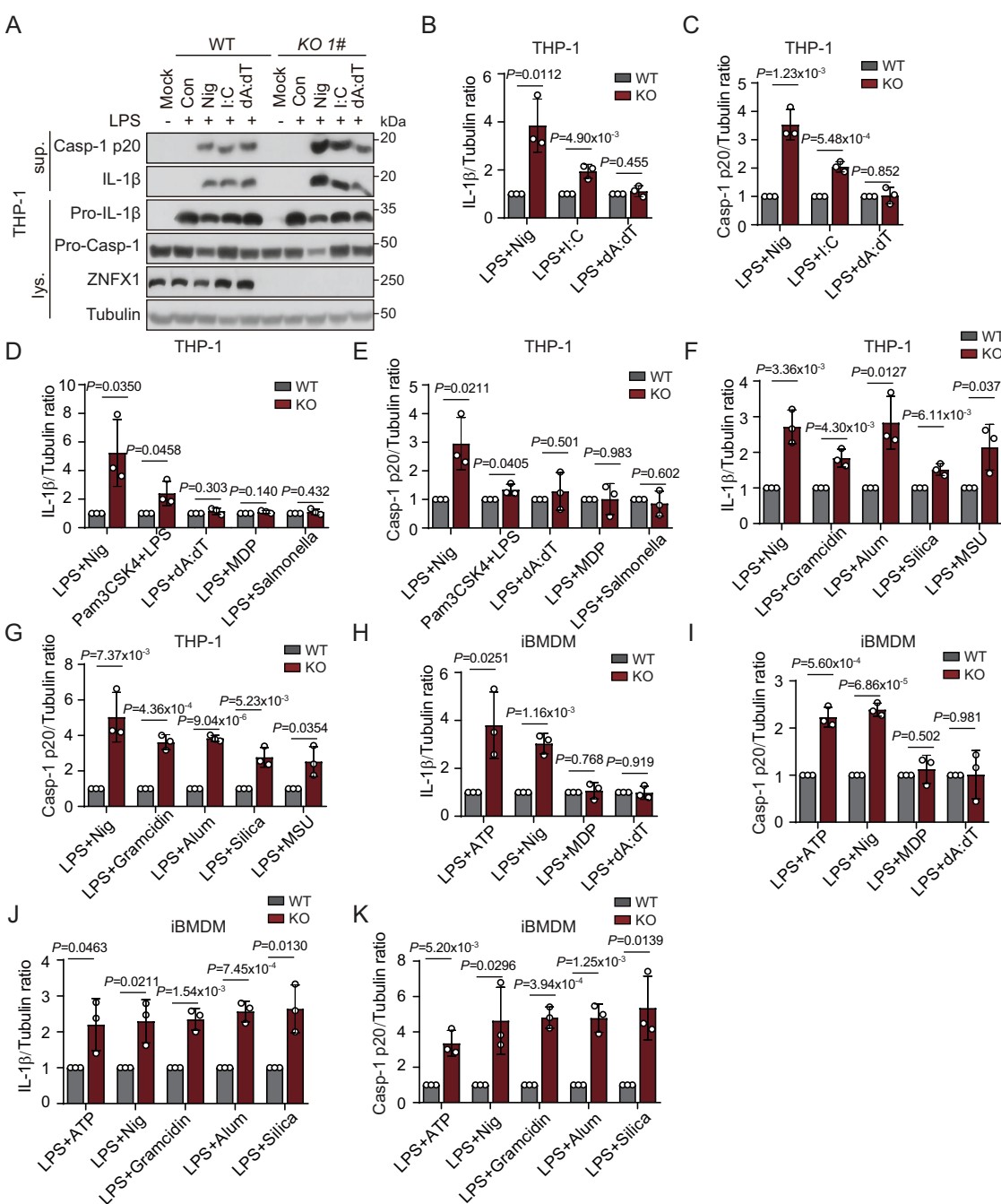

**Figure EV1.  ZNFX1 specifically suppresses the activation of the NLRP3 inflammasome in vitro (related to Fig. 1).**

(A) LPS-primed wildtype (WT) or *ZNFX1* knockout (KO) THP-1-derived macrophages were treated with the indicated inflammation agonists, caspase-1 and IL-1β in the supernatant (sup.) and cell lysate (Lys.) were separated by SDS-PAGE and immunoblotted with the indicated antibodies. Mock represents macrophages primed with PBS without further stimulation. Con control, Nig nigericin, I:C poly(I:C), dA:dT, poly(dA:dT). (B, C) Quantification of IL-1β and caspase-1 P20 protein levels in (A). $n = 3$ biological replicates, error bar, ±s.d. (D–G) Quantification of IL-1β and caspase-1 P20 protein levels in Fig. 1A, B. $n = 3$ biological replicates, error bar ± s.d. (H–K) Quantification of IL-1β and caspase-1 P20 protein levels in Fig. 1C, D. $n = 3$ biological replicates, error bar, ±s.d. Student's *t*-test, two-tailed for (B–K). For each biological replicate, band intensity was measured using ImageJ. The WT control was set to 1, and the ratio for *ZNFX1* KO cells was calculated by dividing their intensity by the corresponding WT control intensity. Source data are available online for this figure.

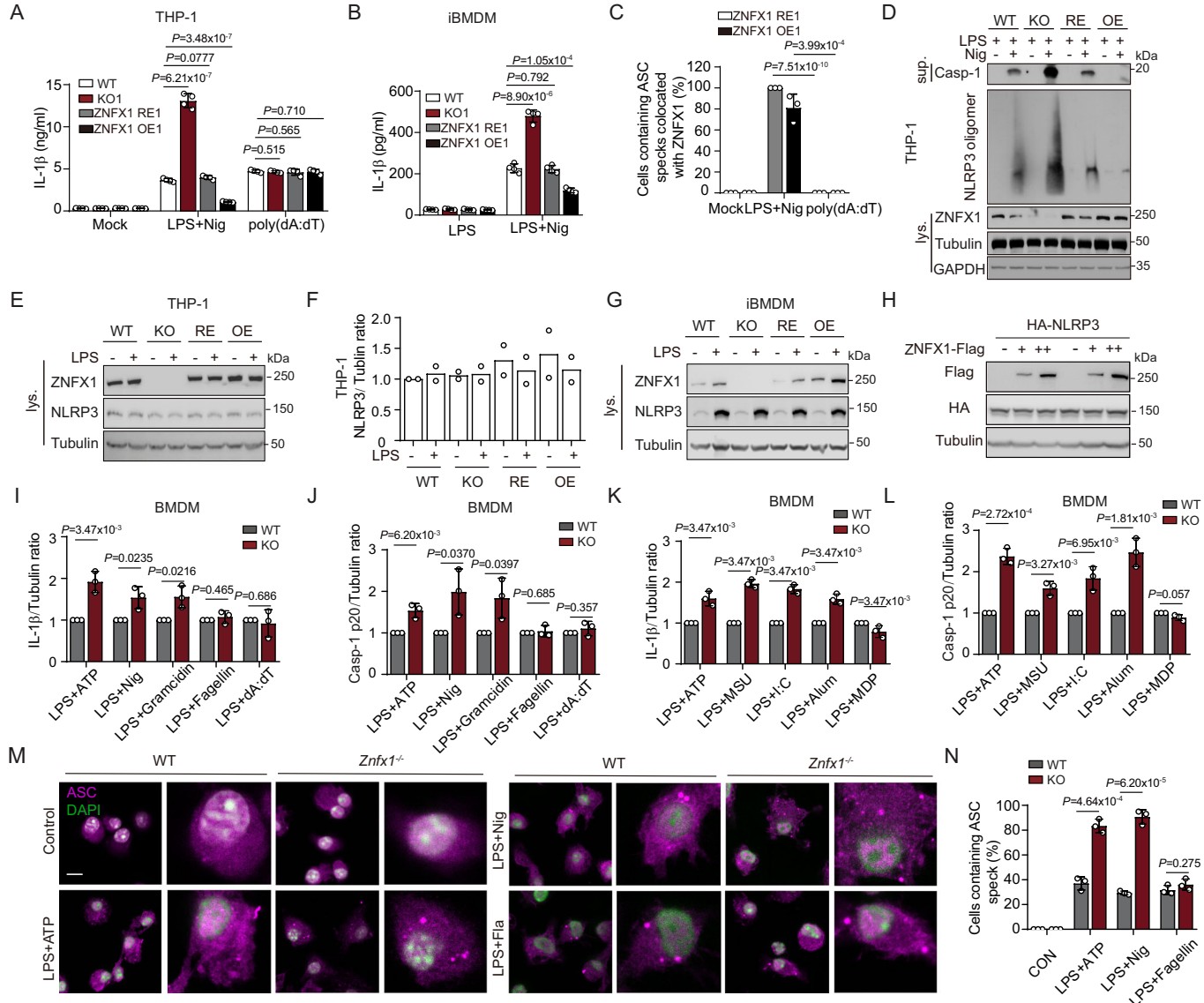

**Figure EV2.  ZNFX1 inhibits the assembly of mature NLRP3 inflammasome.**

(A, B) ELISA detection of IL-1β in the supernatant as shown in Fig. 1G, H. Student's *t*-test, two-tailed. C Quantification of the percentage of cells with ASC specks containing GFP-ZNFX1 as shown in Fig. 1E. Student's *t*-test, two-tailed. (D) SDD-AGE analysis of NLRP3 oligomerization in the lysates of nigericin-treated macrophages. (E, F) Immunoblotting to detect indicated proteins in WT, *ZNFX1* KO, *ZNFX1* rescue (RE), and *ZNFX1* overexpression (OE) THP-1 derived macrophage with or without LPS priming (E). Quantification of NLRP3 protein level (F). n = 2 biological replicates. For each biological replicate, band intensity was measured using ImageJ. The WT control was set to 1, and the ratio for *ZNFX1* KO cells was calculated by dividing their intensity by the corresponding WT control intensity. (G) Immunoblotting to detect indicated proteins in WT, *ZNFX1* KO, *ZNFX1* RE, and *ZNFX1* OE IBMDM cells with or without LPS priming. (H) HeLa cells were transfected with HA-NLRP3 and increased level 3xFLAG-ZNFX1. Immunoblotting was used to detect indicated proteins. (I–L) Quantification of IL-1β and caspase-1 P20 protein level as shown in Fig. 1I, J. n = 3 biological replicates. Error bars represent ± s.d. Student's *t*-test, two-tailed. For each biological replicate, band intensity was measured using ImageJ. The WT control was set to 1, and the ratio for *ZNFX1* KO cells was calculated by dividing their intensity by the corresponding WT control intensity. (M) Primed WT and *Znfx1* BMDM cells were treated with the indicated inflammasome agonist. ASC speck was detected with anti-ASC antibody followed by Alex fluor-568 conjugated secondary antibody. Scale bar, 10 μm. (N) Quantification of the percentage of cells containing ASC specks in (M). n = 3 biological replicates. Error bars represent ± s.d. For each replicate, 100 cells were quantified. Student's *t*-test, two-tailed. Source data are available online for this figure.

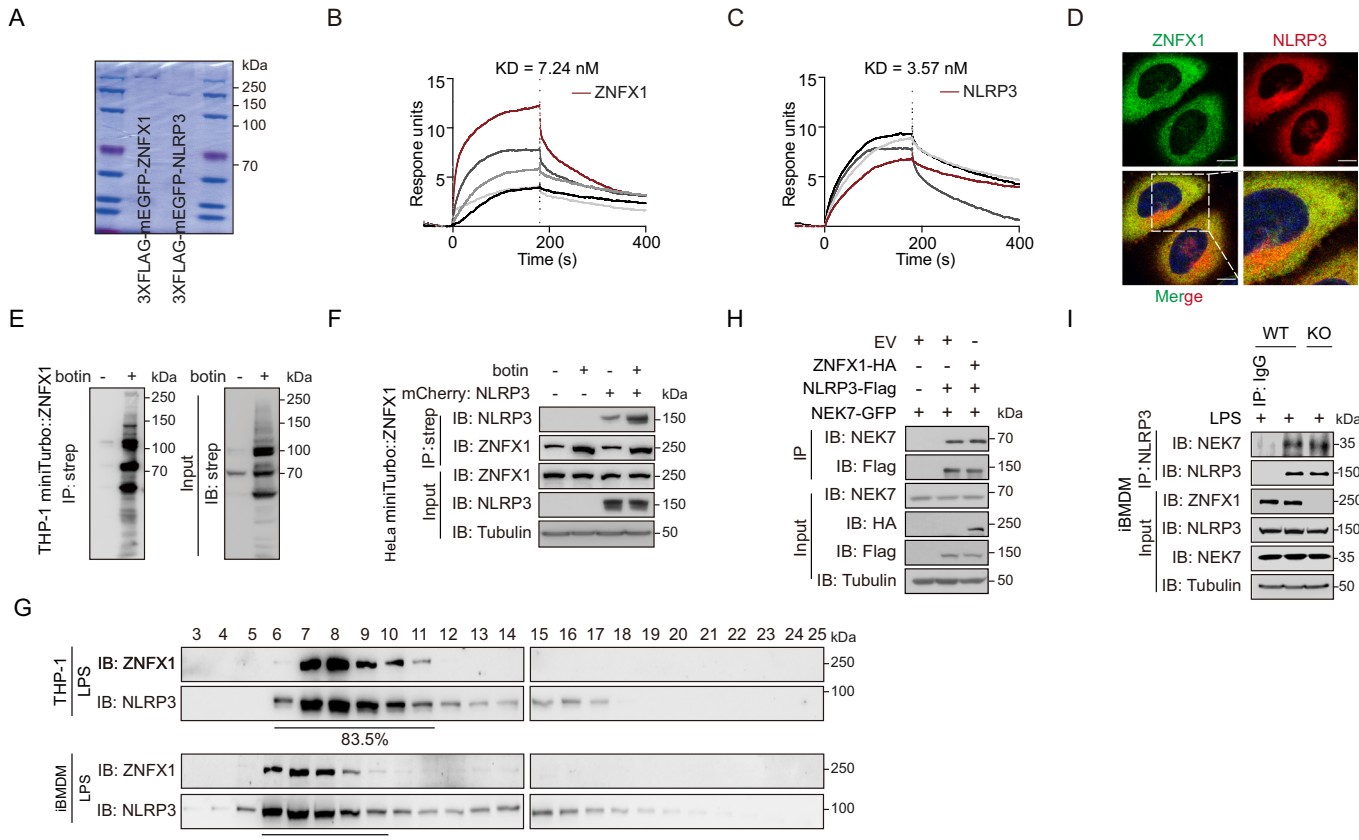

**Figure EV3. ZNFX1 interacts with NLRP3 related to Fig. 3.**

(**A**) Coomassie brilliant blue staining of purified 3xFLAG-GFP-ZNFX1 and 3xFLAG-GFP-NLRP3 proteins. (**B, C**) Purified 3xFLAG-ZNFX1 or 3xFLAG-NLRP3 proteins were immobilized, and proteins associated with the immobilized proteins were analyzed using surface plasmon resonance (SPR). (**D**) Fluorescence imaging of HeLa cells stably expressed GFP-ZNFX1 and mCherry-NLRP3. Scale bar, 10 μm. (**E**) Immunoblot analysis of proteins in cell lysate and miniTurbo-ZNFX1 catalyzed biotinylated species from THP-1 derived macrophages before and after adding biotin substrate using strep-HRP. (**F**) HeLa cells stably expressing miniTurbo-ZNFX1 or miniTurbo-ZNFX1 and mCherry-NLRP3 were treated with biotin to initiate proximity biotin labeling. Proteins from cell lysate and those enriched by streptavidin were resolved by SDS-PAGE and detected by immunoblot. (**G**) Proteins from resting THP-1-derived macrophages or iBMDM were separated using a size-exclusion Column. Eluted protein fractions were detected by immunoblot. (**H**) NEK7-GFP was co-expressed with empty vector (EV), HA-ZNFX1, or FLAG-NLRP3 in HEK293T cells. Coimmunoprecipitation analysis of indicated proteins was performed using α-GFP agarose beads. (**I**) Cell lysate from WT or *Znfx1* KO iBMDM cells was precipitated with NLRP3. Immunoblot analysis of proteins in a cell lysate or co-precipitated fraction using indicated antibodies. Source data are available online for this figure.

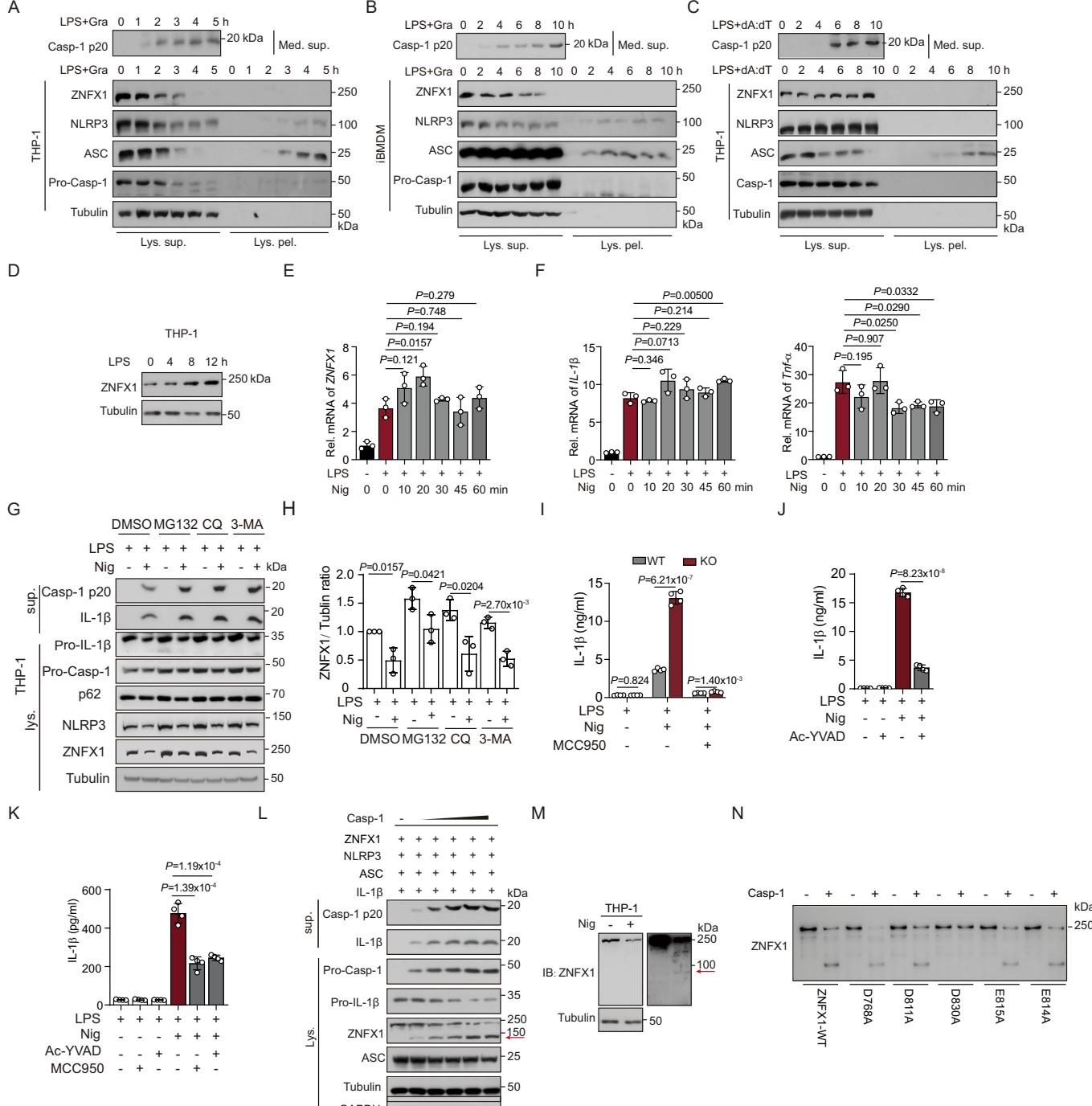

◀

**Figure EV4.  The decreased *ZNFX1* protein level in response to NLRP3 inflammasome activation is largely posttranscriptional and independent of the ubiquitin-proteasome and autophagy pathway.**

(A, B) LPS-primed THP-1-derived macrophages (A) or iBMDM (B) were treated with gramicidin for the indicated time. Proteins from medium supernatant, cell lysate supernatant, and pellet were detected by immunoblot. (C) LPS-primed THP-1 derived macrophages were transfected with poly (dA:dT). Proteins from medium supernatant, cell lysate supernatant, and pellet post-transfection were detected by immunoblot. (D–F) THP-1 derived macrophages were primed with LPS and treated with 10 μM nigericin for the indicated time, the protein level of *ZNFX1* (D), mRNA level of *ZNFX1* (E), and mRNA level of *IL-1β* (F) and *TNF-α* (F) were measured by immunoblotting or quantitative reverse transcription PCR (qRT-PCR). $n = 3$ biological replicates, mean ± s.d., Student's *t*-test, two-tailed. (G, H) LPS-primed THP-1 derived macrophages were pretreated with proteasome inhibitor MG132 (10 μM), autophagy inhibitor 3-MA (20 mM), or CQ (100 μM) for 4 h, and then stimulated with nigericin. Proteins from medium supernatant and cell lysate were detected by immunoblot (G). Quantification of *ZNFX1* protein was performed using ImageJ (H). $n = 3$ biological replicates. Error bars represent ± s.d. Student's *t*-test, two-tailed. For each biological replicate, band intensity was measured using ImageJ. The WT control was set to 1, and the ratio for *ZNFX1* KO cells was calculated by dividing their intensity by the corresponding WT control intensity. (I–K) Primed WT THP-1 derived macrophages (I–J) or iBMDM (K) were subjected to LPS or nigericin treatment with or without MCC950 or caspase-1 inhibitor (Ac-YVAD-cmk) pretreatment. Proteins from culture medium supernatant were detected by ELISA. $n = 3$ biological replicates, mean ± s.d., Student's *t*-test, two-tailed. (L) Indicated proteins were co-expressed in HeLa cells. Proteins from medium supernatant and cell lysate were detected by immunoblot. (M) Proteins from cell lysate of LPS + nigericin-treated THP-1 cells were detected by immunoblot. The red arrow indicates putative cleaved *ZNFX1* fragments. (N) Purified 3xFLAG-mEGFP-ZNFX1 with or without potential caspase-1 cleavage sites mutated were incubated with caspase-1 protein. Products were detected by immunoblot using α-FLAG antibody. This is an independent biological replicate related to Fig. 6D. Source data are available online for this figure.

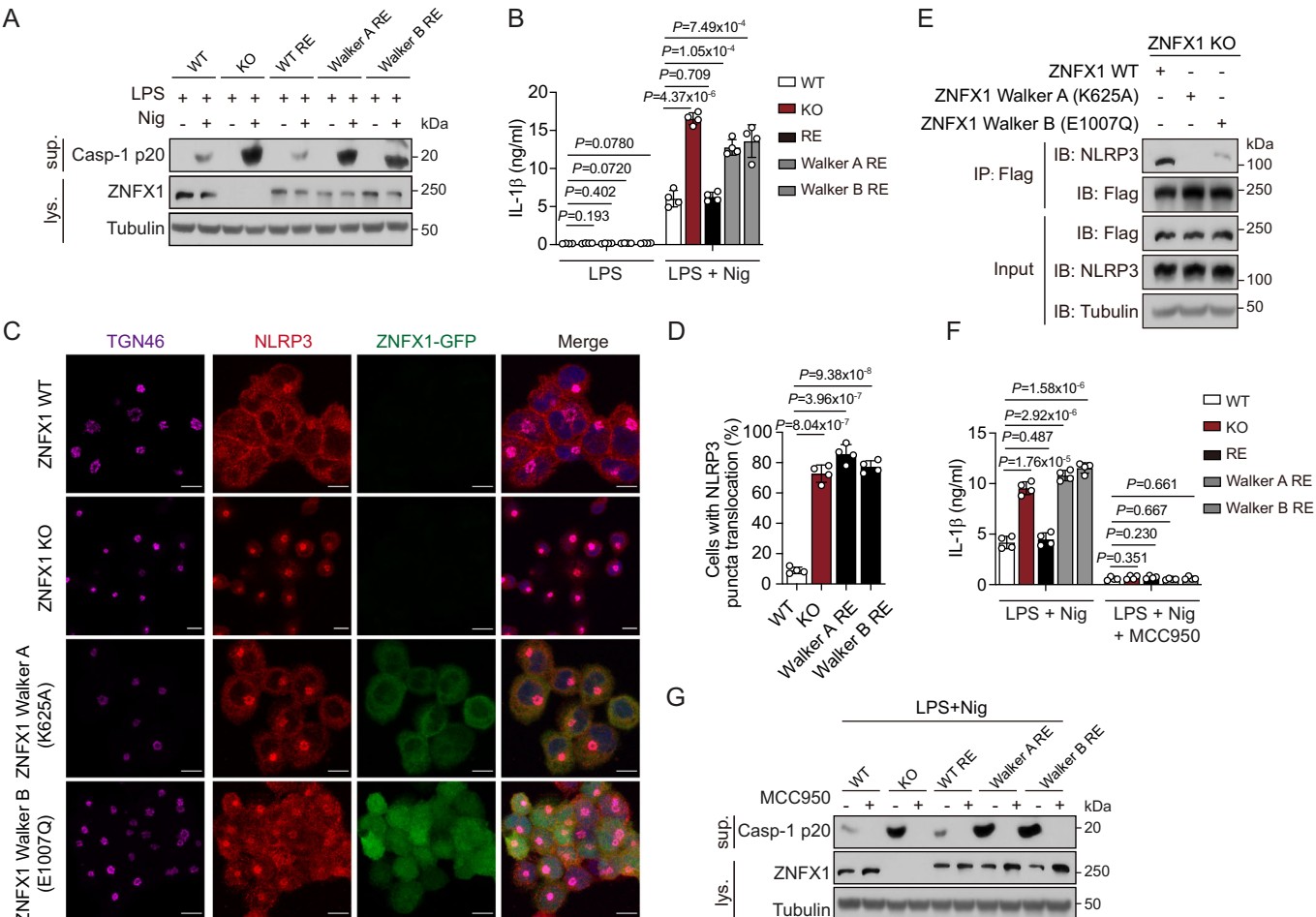

**Figure EV5. The helicase activity of *ZNFX1* is critical for NLRP3 inflammasome inhibition.**

(**A, B**) WT *ZNFX1* or *ZNFX1* containing Walker A or Walker B motif mutations were introduced back to *ZNFX1* KO THP-1 cells by lentivirus-mediated delivery. Cells were primed with LPS and stimulated with nigericin. Proteins from medium supernatant or cell lysate were examined with immunoblot (**A**) and ELISA (**B**). $n = 4$ biological replicates, mean ± s.d., Student's *t*-test, two-tailed. (**C**) WT, *ZNFX1* KO, or *ZNFX1* KO cells complemented with *ZNFX1* harboring helicase mutations were fixed and stained with anti-TGN46 and anti-NLRP3 antibodies, followed by Alex fluor-647 and Alex fluor-568 conjugated secondary antibodies, respectively. Scale bar, 10 μm. (**D**) Quantification of the percentage of cells with NLRP3's TGN translocation from 100 cells in (**C**). $n = 3$, mean ± s.d., two-sided Student's *t*-test. (**E**) Coimmunoprecipitation analysis of NLRP3 with WT or helicase mutants of ZNFX1. Proteins in input cell lysate, as well as precipitation, were detected with indicated antibodies. (**F, G**) Cells with indicated genotype were primed with LPS and treated with nigericin, with or without MCC950 pretreatment, NLRP3 inflammation activation was measured by ELISA in (**F**) and immunoblot in (**G**). $n = 4$ biological replicates, mean ± s.d., Student's *t*-test, two-tailed. Source data are available online for this figure.

