## [Peer Review File · The EMBO Journal]

The human disease-associated gene *ZNFX1* controls inflammation through inhibition of the NLRP3 inflammasome

Jing Huang, Yao Wang, Xin Jia, Changfeng Zhao, Meiqi Zhang, Mi Bao, Pan Fu, Cuiqin Cheng, Ruona Shi, Xiaofei Zhang, Jun Cui, Gang Wan, and Anlong Xu

Corresponding author(s): Gang Wan (wangang5@mail.sysu.edu.cn), Anlong Xu (lssxal@mail.sysu.edu.cn), Yao Wang (yaowang@bucm.edu.cn)

Review Timeline:

Submission Date:	21st Feb 24
Editorial Decision:	5th Apr 24
Revision Received:	4th Jul 24
Editorial Decision:	7th Aug 24
Revision Received:	13th Aug 24
Accepted:	20th Aug 24

Editor: Ioannis Papaioannou

Transaction Report:

Dear Prof. Wan,

Thank you for submitting your manuscript EMBOJ-2024-117038 for consideration by The EMBO Journal. It has now been seen by three experts in the field, and we have received the full set of their comments, which are included below.

As you will see, the referees recognize that this is a novel and interesting study with compelling data supporting some of the main conclusions, but they also identify a number of limitations and point out that further experiments are necessary to strengthen the study with conclusive evidence to fully support all conclusions. Importantly, they mention that some key experiments should be repeated in primary cells from the wild-type and ZNFX1 knockout mice; they raise concerns regarding insufficient support of your proposed model whereby caspase-1 cleaves ZNFX1; and they point out that there are some missing controls and quantification in some experiments. They further provide additional comments and suggestions for the improvement of the manuscript.

Given the referees' comments and recommendations, I would like to invite you to submit a revised version of the manuscript along with a detailed point-by-point response addressing all referees' comments. I should add that it is EMBO Journal policy to allow only a single round of major experimental revision, and acceptance of your manuscript will therefore depend on the completeness of your responses in this revised version. If you have any questions or comments, we can discuss further in a video call, if you like.

We generally allow three months as standard revision time (July 4, 2024). As a matter of policy, competing manuscripts published during this period will not negatively impact our assessment of the conceptual advance presented by your study. However, we request that you contact us as soon as possible upon publication of any related work, to discuss how to proceed. Should you foresee a problem in meeting this three-month deadline, please let us know in advance and we may be able to grant an extension.

Thank you for the opportunity to consider your work for publication in The EMBO Journal. I look forward to your revision.

Yours sincerely,

Instructions for preparing your revised manuscript

1. When you are ready to submit the revision, please upload:

- A Word file of the manuscript text (including legends of main Figures, EV Figures and Tables). Please make sure that changes are highlighted (or "tracked") to be clearly visible.

- Individual production-quality figure files (one file per figure). When assembling your figures, please refer to our figure preparation guidelines in order to ensure proper formatting and readability in print as well as on screen:

If the data shown in a figure are obtained from n {less than or equal to} 2, please use scatter plots showing the individual data points.

- i. the name of the statistical test used to generate error bars and P values
- ii. the number (n) of independent experiments (please specify technical or biological replicates) underlying each data point (discussion of statistical methodology can be reported in the Materials and Methods section, but figure legends should contain a basic description of n , P , and the test applied)
- iii. the nature of the bars and error bars (s.d., s.e.m.).

- A point-by-point response to the referees' comments, with a detailed description of the changes made (as a word file). All referees' concerns must be fully addressed and their suggestions taken on board. When preparing your letter of response to the referees' comments, please bear in mind that this will form part of the Review Process File and will therefore be available online to the community. Please note that you have the possibility to opt out of the transparent process at any stage prior to publication

by letting the editorial office know (contact@embojournal.org); if you do opt out, the Review Process File link will point to the following statement: "No Review Process File is available with this article, as the authors have chosen not to make the review process public in this case.". For more details on our Transparent Editorial Process, please visit our website: <https://www.embopress.org/page/journal/14602075/authorguide#transparentprocess>

- Expanded View (EV) files (replacing Supplementary Information) that are collapsible/expandable online. A maximum of 5 EV Figures can be typeset. EV Figures should be cited as "Figure EV1, Figure EV2" etc. in the text, and their respective legends should be included in the manuscript file after the legends of regular figures. See detailed instructions regarding Expanded View files here:

- For the figures that you do NOT wish to display as Expanded View figures, they should be bundled together with their legends in a single PDF file called "Appendix", which should start with a short Table of Contents (including page numbers). Appendix figures should be referred to in the main text as: "Appendix Figure S1, Appendix Figure S2" etc. Please see detailed instructions here: <https://www.embopress.org/page/journal/14602075/authorguide#expandedview>

- A complete author checklist, which you can download from our author guidelines (<https://www.embopress.org/page/journal/14602075/authorguide>). Please note that the checklist will also be part of the Review Process File.

2. Please note that no statistics should be calculated and shown in Figures if $n=2$.

3. Before submitting your revision, primary datasets (and computer code, where appropriate) produced in this study need to be deposited in appropriate public databases (see <https://www.embopress.org/page/journal/14602075/authorguide#dataavailability>). The accession numbers and databases should be listed in a formal "Data availability" section (placed after Materials and Methods) that follows the model below (see also <https://www.embopress.org/page/journal/14602075/authorguide#dataavailability>):

Data availability

- RNA-seq data: Gene Expression Omnibus GSE46843 (<https://www.ncbi.nlm.nih.gov/geo/query/acc.cgi?acc=GSE46843>)
- [data type]: [name of the resource] [accession number/identifier/doi] ([URL or identifiers.org/DATABASE:ACCESSION])

*** All links should resolve to a page where the data can be accessed. ***

*** Please remember to provide in the Data availability section of your revised manuscript reviewer passwords if the datasets are not yet public. ***

*** The Data Availability Section is restricted to new primary data that are part of this study. In case you have no data that require deposition in a public database, please state so instead of referring to the database: "Our study includes no data deposited in public repositories." under the heading "Data availability". ***

4. Please check that the title and the abstract of the manuscript are brief, yet explicit, even to non-specialists. The length of the title should not exceed 100 characters, and the abstract should be a single paragraph not exceeding 175 words.

5. Please also note our reference format: <https://www.embopress.org/page/journal/14602075/authorguide#referencesformat>.

7. Please remember: digital image enhancement is acceptable practice, as long as it accurately represents the original data and conforms to community standards. If a figure has been subjected to significant electronic manipulation, this must be noted in the figure legend or in the "Materials and Methods" section. The editors reserve the right to request original versions of figures and the original images that were used to assemble the figure.

8. Our journal encourages inclusion of data citations in the reference list to directly cite datasets that were obtained from public databases. Data citations in the article text are distinct from normal bibliographical citations and should directly link to the database records from which the data can be accessed. In the main text, data citations are formatted as follows: "Data ref: Smith et al, 2001" or "Data ref: NCBI Sequence Read Archive PRJNA342805, 2017". In the Reference list, data citations must be labeled with "[DATASET]". A data reference must provide the database name, accession number/identifiers, and a resolvable link to the landing page from which the data can be accessed at the end of the reference. Further instructions are available at:

<https://www.embopress.org/page/journal/14602075/authorguide#referencesformat>.

9. We request authors to consider both actual and perceived competing interests. Please review our policy (<https://www.embopress.org/page/journal/14602075/authorguide#conflictsofinterest>) and update your competing interests statement if necessary. Please name this section 'Disclosure and competing interests statement' and place it after the Acknowledgements section.

10. Please note that all corresponding authors are required to provide an ORCID ID upon submission of a revised manuscript (<https://orcid.org/>). Please find instructions on how to link your ORCID ID to your account in our manuscript tracking system in our Author guidelines (<https://www.embopress.org/page/journal/14602075/authorguide#authorshipguidelines>).

11. We use CRediT to specify the contributions of each author in the journal submission system. CRediT replaces the author contribution section, which should be removed from the manuscript. Please use the free text box to provide more detailed descriptions. See also guide to authors: <https://www.embopress.org/page/journal/14602075/authorguide#authorshipguidelines>.

13. We would also welcome the submission of cover suggestions or motifs to be used by our Graphics Illustrator in designing a cover.

14. Please use the link below to submit your revision:
<https://emboj.msubmit.net/cgi-bin/main.plex>

Referee #1:

Overall, I found this to be an interesting study with compelling evidence supporting some of the key observations. The experiments are well controlled with both loss of function and reconstitution experiments performed across a range of assays and cell types. The overall model that ZNFX1 interacts with the NLRP3 inflammasome to restrain its activation is supported by the findings both in mouse iBMDM, Thp1 and in vivo. The structure functions studies map the interaction of ZNFX1 and NLRP1 and the consequences of these interactions on inflammasome dependent processes. ZNFX1 retains NLRP3 in the cytoplasm and hinders the accumulation of NLRP3 on the Trans-Golgi network (TGN) in the resting state to prevent its activation. Furthermore, the findings that NLRP3 dependent caspase-1 activation leads to cleavage of ZNFX1 as a mechanism to amplify downstream signaling is also supported by the data. Lastly, the authors demonstrate that while wild-type ZNFX1, restrains NLRP3 inflammasome responses the disease associated mutations do not.

The following key issues would need to be addressed:

1. Several of the key experiments should be performed in primary cells from the WT and ZNFX1-KO mice. Bone marrow derived macrophages or peritoneal macrophages should be evaluated since the mice are available and this would be more compelling than the cell line studies. The studies in figure 1 using cell lines should be performed in primary cells.
2. The authors propose a model whereby ZNFX1 can be cleaved by caspase-1 in response to NLRP3 agonists, but not agonists of the AIM2, NLRP1b, or NLRC4 inflammasome, suggesting a specific cleavage mechanism. Since caspase1 activation is common to other stimuli the authors should provide a molecular explanation for these findings.

Referee #2:

ZNFX-1 is a conserved helicase which has been implicated as a regulator of innate immunity in mammals. This paper reports on a novel role for ZNFX-1 as an inhibitor of inflammasome assembly. The authors propose a model whereby ZNFX-1 limits inflammasome activation by directly binding to inflammasome protein NLRP3 in the cytoplasm, which prevents NLRP3 translocation to TGN vesicles. The authors also propose that, in turn, inflammasome-activated caspase I cleaves ZNFX-1, thus turning off the inhibitory effects of ZNFX-1 to allow for full on inflammasome activation. The idea that ZNFX-1 serves as a natural break for the inflammasome provides a possible explanation for why patients with ZNFX-1 mutations are susceptible to hyper-inflammation.

This paper and the figures are very dense. In several cases, the order of panels in the Figures does not seem to follow the order in the text. Most western blot results are not quantified - making it difficult to assess reproducibility. The conclusion that ZNFX-1 null cells hyperactivate NLRP3-dependent responses appears solid. The conclusion that ZNFX-1 is targeted and inactivated by Caspase 1 and/or stress granules, however, are not as well supported by the evidence. These experiments seem to detract from the main message that ZNFX-1 limits inflammasome activation; the authors may want to consider removing those experiments to maintain the focus on the main conclusion.

Specific comments:

1. The main and most important result of this paper is that ZNFX-1 knock out cells exhibit increased production of IL-1beta and cleaved caspase 1 upon activation of the inflammasome - The authors refer to many panels but provide no systematic quantification to support this result.
2. In a prior paper, the authors suggested that ZNFX-1 localizes to mitochondrial surface <https://pubmed.ncbi.nlm.nih.gov/31685995/>. In this paper, they claim that ZNFX-1 is cytoplasmic???
3. Fig. 3 is very confusing - Line 240 in text seems to come prematurely because fig. 3C has not been discussed yet. The decrease of ZNFX-1 full length is shown where exactly ?? Fig. 3F??? please make it easier for reader to follow. Also quantification of this decrease should be provided.
4. There appears to be no direct evidence that caspase-1 cleaves ZNFX1 in vivo and that this in turns serves as a "a trigger for a feed-forward loop that enhances NLRP3 inflammasome activation" as stated in text. The main evidence is shown in Fig. 4E where a caspase inhibitor appears to suppress loss ZNFX1 but this is just one data point with no quantification. It would seem that to make this claim would require at a minimum mutating the caspase sites in ZNFX1 in cells and examining the effect on NLRP3 activation???
5. Where is the quantification to show that addition of ZNFX1 increases G3BP condensation in vitro? Can the authors exclude an RNA contaminant? These experiments appear to detract from the main point of the paper.
6. What is the evidence for a direct inhibition of ZNFX1 by G3BP/stress granules? To demonstrate in vivo relevance, the authors would need to show at a minimum that G3BP null cells exhibit increased ZNFX1 activity?? These experiments appear to detract from the main point of the paper.

Fig. 6B is not convincing - there is no quantification to support the view that mutant ZNFX-1 are defective

Fig. 6C needs to show WT rescue control

Please provide figure numbers on the figure pages.

Referee #3:

In this paper, Huang et al report that ZNFX1, as an ISG and dsRNA virus sensor, selectively suppress activation of NLRP3 inflammasome to control inflammation. Mechanistically, ZNFX1 interacts with NLRP3 and sequesters NLRP3 away from Trans-Golgi network (TGN) during the priming stage. ZNFX1 could be cleaved by caspase-1, resulting in a feedback regulatory mechanism for NLRP3 inflammasome activation. Additionally, ZNFX1 facilitates stress granule formation in a caspase-1-independent manner during activation stage. The study uncovers another mechanism that retains NLRP3 inactive in resting state, and a novel role of ZNFX1 in controlling inflammation. Overall, the findings are novel and interesting. However, the following concerns and questions should be addressed to further strengthen the study.

Major comments:

1. The authors show that ZNFX1 promoted stress granules formation and the aggregation of ZNFX1 in stress granules could enhance NLRP3 inflammasome activation during the activation stage. Whether the role is dependent of stress granules, and the authors may explore it using inhibitor of stress granules formation, such as emetine (DOI: 10.1083/jcb.200502088, DOI: 10.1038/s41586-023-06726-w).
- 2 NLRP3 protein level is a key limiting step for inflammasome activation (DOI: 10.3109/10409238.2012.694844, DOI: 10.1038/ncomms13727). Whether ZNFX1 could affect NLRP3 expression should be investigated.
3. Figure 1N showed that during LPS injection for 6 days, none of the mice in WT group (Znfx1+/+) has died, which is confusing. The authors should explain it.
4. Figure EV 1B & G, the data showed the expression of TNF- α exist great discrepancy between LPS treatment and LPS+stimuli, which seemed not in accordance with previous study (DOI: 10.1038/ncomms13727, DOI: 10.1016/j.celrep.2024.113752). The authors should explain or further confirm this issue.

5. Figure3A, the authors should perform a live cell imaging timeline of the indicated cells in order to display the location changes of NLRP3, ZNFX1 and TNG46.

Minor comments:

1. In the Abstract section, the description related stress granules or virus should be included.
2. The abbreviations are usually used defined at the first use in the abstract as well as in the main text, such as ZNFX1 in the abstract.
3. Figure EV5I, the data could not support the corresponding conclusion well (line295).
4. The expression of GAPDH or Tubulin is too weak, such as Figure 4A, C.
5. There are numerous grammatical mistakes in the manuscript (such as Line197).

First and foremost, I want to thank you all for dedicating your time and effort to review our manuscript. Your feedback has enhanced the quality of our study and provided valuable guidance for future research endeavors.

In response to reviewer comments and suggestions, we have conducted additional experiments and made text and conclusion modifications to provide a more precise and rigorous description of our findings. To facilitate your ability to assess changes, we have highlighted the changes we have made in blue in the revised manuscript as well as in our response to reviewer concerns below. Textual changes to the manuscript are included below in italicized blue text.

Below is an outline of the major changes we have made to the revised manuscript as well as the major conclusions we make in this paper.

Major changes made in the revised manuscript.

- 1) We have reorganized Figure 1 from our initial submission into Figures 1 and 2, and Figure 4 from our initial submission into Figures 5 and 6 in our revised manuscript. We also reordered panels in Figure 3 (now Figure 4 in our revised manuscript) to enhance flow and clarity. Additionally, the EV figures have been divided into Figures EV1-EV5 and Appendix Figures S1-S5. We believe these changes better convey our information.
- 2) We have provided extensive quantifications for our western blot results from biological replicates (Figures 5E-F, 5H, 6G-I, 7C, EV1B-K, EV2I-L, EV4H, and Appendix Fig. S3D).
- 3) We have included further data showing that the level of NLRP3 inflammasome activation is increased in primary *Znfx1*^{-/-} BMDM cells compared with WT (Figs. 1J, EV2M-N).
- 4) We found that ZNFX1(D830A) suppresses the activation of the NLRP3 inflammasome and the cleavage of caspase-1, supporting the idea that a feed-forward loop further enhances NLRP3 inflammasome activation (Figs. 6E-I).
- 5) We have removed data related to poly(I:C), EMCV, and stress granules for the following reasons: first, we cannot provide direct evidence that suppressing stress granules enhances NLRP3 inflammasome activation; second, as pointed out by reviewer #2, these data largely distract from the main focus of the paper.

Major findings in the revised manuscript.

- 1) We identify that ZNFX1 specifically suppresses the activation of the NLRP3 inflammasome *in vitro*.
- 2) We provide evidence supporting the inhibition of the NLRP3 inflammasome by ZNFX1 *in vivo*.
- 3) We find that ZNFX1 likely interacts directly with NLRP3.
- 4) We show that ZNFX1 prevents the translocation of NLRP3 to TGN38+/TGN46+ vesicles.
- 5) We find that the decrease in ZNFX1 protein levels during NLRP3 inflammasome activation is suppressed by a Caspase-1 inhibitor but not by proteasome or autophagy pathway inhibitors.
- 6) We show that Caspase-1 cleaves ZNFX1 both *in vitro* and *in vivo*, and identify an important cleavage site in ZNFX1 (D830). ZNFX1(D830A) suppresses the activation of the NLRP3 inflammasome and cleavage of Caspase-1, supporting the idea that a feed-forward loop further enhances NLRP3 inflammasome activation.
- 7) We link pathogenic lesions in ZNFX1 with hyperactivation of the NLRP3 inflammasome, providing potential insights into the pathogenesis and treatment of this genetic disease.

Referee #1:

Overall, I found this to be an interesting study with compelling evidence supporting some of the key observations. The experiments are well controlled with both loss of function and reconstitution experiments performed across a range of assays and cell types. The overall model that ZNFX1 interacts with the NLRP3 inflammasome to restrain its activation is supported by the findings both in mouse iBMDM, Thp1 and *in vivo*. The structure functions studies map the interaction of ZNFX1 and NLRP1 and the consequences of these interactions on inflammasome dependent processes. ZNFX1 retains NLRP3 in the cytoplasm and hinders the accumulation of NLRP3 on the Trans-Golgi network (TGN) in the resting state to prevent its activation. Furthermore, the findings that NLRP3 dependent caspase-1 activation leads to cleavage of ZNFX1 as a mechanism to amplify downstream signaling is also supported by the data. Lastly, the authors demonstrate that while wild-type ZNFX1, restrains NLRP3 inflammasome responses the disease associated mutations do not.

We thank the reviewer for the kind words.

The following key issues would need to be addressed:

1. Several of the key experiments should be performed in primary cells from the WT and ZNFX1-KO mice. Bone marrow derived macrophages or peritoneal macrophages should be evaluated since the mice are available and this would be more compelling

than the cell line studies. The studies in figure 1 using cell lines should be performed in primary cells.

We would like to thank the reviewer for this excellent suggestion. We repeated the experiments conducted in THP-1 and iBMDM cells from Figure 1 in primary BMDM cells obtained from WT and *Znfx1*^{-/-} mice. Specifically, we provided additional evidence showing that NLRP3 inflammasome activation levels, but not other inflammasome activation levels, are higher in *Znfx1*^{-/-} BMDM cells compared to WT BMDM cells (Fig. 1J, EV2I-L). Additionally, the assembly of ASC specks is increased in *Znfx1*^{-/-} BMDM cells compared to WT BMDM cells (Fig. EV2M-N).

Results (Lines 150-154):

“To further validate our results, we obtained primary BMDMs from WT and Znfx1^{-/-} mice and confirmed that NLRP3 inflammasome activation was increased (1.5- to 2.5-fold) in Znfx1^{-/-} BMDMs upon NLRP3 agonist treatment but not upon NLRC4, AIM2, or NLRP1 agonist treatment (Figs. 1I-K, EV2I-N, and Appendix Fig. S2I).”

2. The authors propose a model whereby ZNFX1 can be cleaved by caspase-1 in response to NLRP3 agonists, but not agonists of the AIM2, NLRP1b, or NLRC4 inflammasome, suggesting a specific cleavage mechanism. Since caspase1 activation is common to other stimuli the authors should provide a molecular explanation for these findings.

Thanks for the great suggestion. We subjected GFP-ZNFX1 reconstituted THP-1 cells to NLRP3, AIM2, NLRP1b, and NLRC4 agonists treatment, respectively. IF results showed that GFP-ZNFX1 colocalizes with ASC speck after NLRP3 inflammasome activation but not AIM2, NLRP1b, and NLRC4 inflammasome activation. This result hints that ZNFX1 may be recruited to mature NLRP3 inflammasome for unknown reason, whereby caspase-1 cleaves ZNFX1 to reduce ZNFX1 level to further release its inhibitory role on NLRP3 inflammasome.

Results (Lines 275-280):

“Consistent with these findings, ZNFX1 localized to ASC specks specifically when the NLRP3 inflammation, but not the AIM2, NLRP1b, or NLRC4 inflammasome, was activated in THP-1 macrophages (Figs. 1E, 4G-H, and EV2C). This suggests that as the ZNFX1 might be recruited to the activated NLRP3 inflammasome, likely exerting a direct inhibitory effect on it.”

Referee #2:

ZNFX-1 is a conserved helicase which has been implicated as a regulator of innate immunity in mammals. This paper reports on a novel role for ZNFX-1 as an inhibitor of inflammasome assembly. The authors propose a model whereby ZNFX-1 limits inflammasome activation by directly binding to inflammasome protein NLRP3 in the cytoplasm, which prevents NLRP3 translocation to TGN vesicles. The authors also propose that, in turn, inflammasome-activated caspase 1 cleaves ZNFX-1, thus turning off the inhibitory effects of ZNFX-1 to allow for full on inflammasome activation. The idea that ZNFX-1 serves as a natural break for the inflammasome provides a possible explanation for why patients with ZNFX-1 mutations are susceptible to hyper-inflammation.

This paper and the figures are very dense. In several cases, the order of panels in the Figures does not seem to follow the order in the text. Most western blot results are not quantified - making it difficult to assess reproducibility. The conclusion that ZNFX-1 null cells hyperactivate NLRP3-dependent responses appears solid. The conclusion that ZNFX-1 is targeted and inactivated by Caspase 1 and/or stress granules, however, are not as well supported by the evidence. These experiments seem to detract from the main message that ZNFX-1 limits inflammasome activation; the authors may want to consider removing those experiments to maintain the focus on the main conclusion.

We are grateful for your critical assessment of our study and your specific suggestions for improvement, as they have been crucial in enhancing the quality of our manuscript and guiding future research. To address these concerns, we have made the following changes:

1) We have divided several figures to make them more concise (specifically, Figure 1 from our initial submission into Figures 1 and 2, and Figure 4 from our initial submission into Figures 5 and 6 in our revised manuscript) and ensured the order of panels in the figures matches the order in the text, facilitating the readability of the manuscript.

2) We have conducted the requested quantification of the western blots from biological replicates.

3) We have removed the stress granule-related data in the revised manuscript, as we agree with the reviewer that these data largely detract from the main focus of the paper.

Please see the details below. We hope you agree that the quality of the manuscript has improved with these changes.

Specific comments:

1. The main and most important result of this paper is that ZNFX-1 knock out cells exhibit increased production of IL-1 β and cleaved caspase 1 upon activation of the

inflammasome – The authors refer to many panels but provide no systematic quantification to support this result.

We thank the reviewer for this excellent suggestion. We quantified the relative levels of mature IL-1 β and Caspase-1 p20 in the supernatants of THP-1-derived macrophages, iBMDM, and primary BMDM cells treated with different inflammasome agonists using ImageJ for all biological replicates. This analysis allows us to more accurately describe the results. We have included this data as Figures EV1B-K and EV2I-L in our revised manuscript.

Results (Lines 109-112):

*“Upon exposure of ZNFX1 KO macrophages to the NLRP3 activator nigericin, we observed a **2 to 5-fold** increase in the production of IL-1 β and cleaved caspase-1 (Figs. 1A-D, EV1A-K, and Appendix Figs. S1A-E).”*

Results (Lines 116-120):

*“To validate the specificity of ZNFX1 in inhibiting the NLRP3 inflammasome, we stimulated both wild-type (WT) and ZNFX1 KO macrophages with various other NLRP3 agonists, and increased levels (**1.5- to 5-fold**) of IL-1 β secretion and caspase-1 cleavage were observed in ZNFX1 KO macrophages (Figs. 1A-D, EV1A-K, and Appendix Figs. S1A-F, S2A-F).”*

Results (Lines 150-154):

*“To further validate our results, we obtained primary BMDMs from WT and Znf1-/- mice and confirmed that NLRP3 inflammasome activation was increased (**1.5- to 2.5-fold**) in Znf1-/- BMDMs upon NLRP3 agonist treatment but not upon NLRC4 or AIM2 agonist treatment (Figs. 1I-K, EV2I-N, and Appendix Fig. S2I).”*

2. In a prior paper, the authors suggested that ZNFX-1 localizes to mitochondrial surface <https://pubmed.ncbi.nlm.nih.gov/31685995/>. In this paper, they claim that ZNFX-1 is cytoplasmic???

Thank you for pointing this out. In our previous work¹, we focused on the role of ZNFX1 in combating viral infections by initiating type I IFN through its interaction with MAVS at the mitochondrial surface. However, this work does not rule out the cytoplasmic localization of ZNFX1. The cytoplasmic presence of ZNFX1 has also been confirmed in HeLa, SV40, and THP-1 cells in a recent study². Together with our findings in this study, these results suggest that there are at least two species of ZNFX1: one at the mitochondrial surface and one in the cytoplasm. Thus, ZNFX1 may reside in different cellular localizations, playing distinct roles.

3. Fig. 3 is very confusing - Line 240 in text seems to come prematurely because fig. 3C has not been discussed yet. The decrease of ZNFX-1 full length is shown where exactly ?? Fig. 3F??? please make it easier for reader to follow. Also quantification of this decrease should be provided.

We apologize for the confusion in our original description of this figure in our initial submission. In our revised manuscript, we have reorganized the panels (now Figure 4, and Figure 3C in our first submission have been removed in our revised manuscript) and refined the description to improve clarity and flow. The confirmation of the decrease of ZNFX1 protein level with quantification in *ZNFX1* KO HeLa cells has been included in the revision (Appendix Figs 3C-D). We hope the reviewer would agree that the flow of our Figures and text are now improved and easier to follow.

Results (lines 248-283):

*“To investigate the dynamics of NLRP3 translocation, we treated HeLa cells stably expressing GFP-ZNFX1 and mCherry-NLRP3 with nigericin. Time-lapse imaging revealed that mCherry-NLRP3 translocated to dispersed TGN46+ vesicles, while GFP-ZNFX1 predominantly remained diffuse (Fig. 4A-B and movie EV1). Immunofluorescence analysis using an anti- α -TGN46 antibody revealed that NLRP3 translocated to the TGN in approximately 10% of the resting macrophages derived from THP-1 cells (Fig. 4C-D). However, the degree of translocation increased significantly to approximately 75% in *ZNFX1* KO cells (Fig. 4C-D). Conversely, the percentage of *ZNFX1* RE cells exhibiting NLRP3 translocation was reduced to approximately 10%, and the percentage of *ZNFX1* OE cells exhibiting NLRP3 translocation was further reduced to less than 5% (Fig. 4C-D). These results were confirmed using immunofluorescence with an anti- α -TGN38 antibody (Appendix Fig. S3A-B). The observed NLRP3 species likely corresponded NLRP3 that formed cage-like structures bound to the TGN and were poised for further activation (Andreeva et al., 2021; Chen & Chen, 2018). These findings suggest that *ZNFX1* retains NLRP3 in the cytoplasm and prevents its translocation to membrane vesicles in the resting state.*

*Previous studies have suggested that HeLa cells do not express ASC (Bauernfeind et al, 2009; Fernandes-Alnemri et al, 2009). The observation that NLRP3 translocated to TGN46+/TGN38+ membrane vesicles while *ZNFX1* remained diffuse in the cytoplasm in response to nigericin treatment suggests that an event occurred before the formation of ASC specks (Fig. 4A-B and movie EV1). When GFP-ZNFX1, mCherry-NLRP3, and mMaroon1-ASC were co-expressed in HeLa cells, they formed large aggregates resembling ASC specks (Figs. 4E-F and Appendix Figs. S3C-D). Consistent with these findings, *ZNFX1* localized to ASC specks specifically when the NLRP3 inflammation, but not the AIM2, NLRP1b, or NLRC4 inflammasome, was activated in THP-1 macrophages (Figs. 1E, 4G-H, and EV2C). This suggests that *ZNFX1* might be recruited to the activated NLRP3 inflammasome, likely exerting a direct inhibitory effect on it. Collectively, these findings indicate that *ZNFX1* plays a role in retaining smaller NLRP3 species in the cytoplasm, preventing their translocation to the TGN and the subsequent formation of larger cage-like structures.”*

4. There appears to be no direct evidence that caspase-1 cleaves ZNFX1 in vivo and that this in turns serves as a "a trigger for a feed-forward loop that enhances NLRP3

inflammasome activation" as stated in text. The main evidence is shown in Fig. 4E where a caspase inhibitor appears to suppress loss ZNFX1 but this is just one data point with no quantification. It would seem that to make this claim would require at a minimum mutating the caspase sites in ZNFX1 in cells and examining the effect on NLRP3 activation???

We thank the reviewer for this excellent suggestion. We introduced ZNFX1(D830A) into ZNFX1 KO THP-1 cells and subjected WT, ZNFX1 KO, ZNFX1 RE, and ZNFX1(D830A) cells to LPS or LPS + Nigericin treatment. The results showed that NLRP3 activation levels in ZNFX1(D830A) cells are lower than in WT or ZNFX1 RE cells. Additionally, ZNFX1 levels decreased during NLRP3 inflammasome activation in WT and ZNFX1 RE cells, but not in ZNFX1(D830A) cells. In addition, We have quantified ZNFX1 levels with or without NLRP3 inflammasome and caspase-1 inhibitors from three biological replicates (Figs. 5E-F and 5H in our revised manuscript).

Results (Lines 358-366):

"We hypothesize that caspase-1-mediated cleavage of ZNFX1 is important for further activation of the NLRP3 inflammasome. To test this hypothesis, we induced the expression of ZNFX1(D830A) in ZNFX1 KO THP-1 cells. LPS-primed WT, ZNFX1 KO, ZNFX1 RE, and ZNFX1(D830A) THP-1 cells were exposed to nigericin (Fig. 6E). Immunoblotting and ELISA showed that the levels of IL-1 β and caspase-1 P20 were significantly lower in ZNFX1(D830A) cells than in WT and ZNFX1 RE cells (Fig. 6E-H). Additionally, the ZNFX1 protein level decreased in WT and ZNFX1 RE cells but not in ZNFX1(D830A) cells upon nigericin treatment (Fig. 6E, 6I)."

5. Where is the quantification to show that addition of ZNFX1 increases G3BP condensation in vitro? Can the authors exclude an RNA contaminant? These experiments appear to detract from the main point of the paper.

Given the comments from all reviewers, we have removed the stress granules-centric section from our revised manuscript. However, we conducted the experiments suggested by reviewer #2 (see below). For instance, RNase A was added during the purification of ZNFX1 and G3BP1, as well as in the phase separation experiment. These results support that ZNFX1 increases G3BP1 phase separation *in vitro*, even without RNA. If reviewer #2 would like a complete description of these results, we would be happy to provide them. However, since these experiments are not part of our revised manuscript, we chose not to include a description of most of this data here.

6. What is the evidence for a direct inhibition of ZNFX1 by G3BP/stress granules? To demonstrate in vivo relevance, the authors would need to show at a minimum that G3BP null cells exhibit increased ZNFX1 activity?? These experiments appear to detract from the main point of the paper.

We agree with the reviewer that this section may distract from the main focus of the paper. We attempted to delete G3BP1 and G3BP2 to create G3BP-null THP-1 cells but were unsuccessful. We also used emetine, a stress granule inhibitor, to prevent stress granule formation, as suggested by reviewer 3#. However, emetine can trigger inflammasome activation even without NLRP3 inflammasome agonist treatment (see below in response to reviewer 3#). After extensive evaluation of existing and newly acquired data, we have decided to remove the stress granules-related data from our revised manuscript.

Fig. 6B is not convincing - there is no quantification to support the view that mutant ZNFX-1 are defective

We have quantified the protein level of the caspase-1 P20/Tubulin ratio from three biological replicates using ImageJ. This data is included as Figure 7C in our revised manuscript.

Fig. 6C needs to show WT rescue control

The requested WT rescue control, along with the quantification, is included as Figure 7D and 7E in our revised manuscript.

Please provide figure numbers on the figure pages.

Thank you for noticing this. Figure numbers have been added in the revision.

Referee #3:

In this paper, Huang et al report that ZNFX1, as an ISG and dsRNA virus sensor, selectively suppress activation of NLRP3 inflammasome to control inflammation. Mechanistically, ZNFX1 interacts with NLRP3 and sequesters NLRP3 away from Trans-Golgi network (TGN) during the priming stage. ZNFX1 could be cleaved by caspase-1, resulting in a feedback regulatory mechanism for NLRP3 inflammasome activation. Additionally, ZNFX1 facilitates stress granule formation in a caspase-1-independent manner during activation stage. The study uncovers another mechanism that retains NLRP3 inactive in resting state, and a novel role of ZNFX1 in controlling inflammation. Overall, the findings are novel and interesting. However, the following concerns and questions should be addressed to further strengthen the study.

Thank you for the positive assessment of our study.

Major comments:

1. The authors show that ZNFX1 promoted stress granules formation and the aggregation of ZNFX1 in stress granules could enhance NLRP3 inflammasome activation during the activation stage. Whether the role is dependent of stress granules, and the authors may explore it using inhibitor of stress granules formation, such as emetine (DOI: 10.1083/jcb.200502088, DOI: 10.1038/s41586-023-06726-w).

We would like to thank the reviewer for the excellent suggestion. We performed the requested experiments by treating THP-1 cells with sodium arsenate, with or without emetine pretreatment. We found that sodium arsenate induces the formation of ZNFX1-containing stress granules, which is inhibited by emetine, consistent with previous reports^{3,4}. Next, we treated THP-1 cells with poly(I:C), with or without emetine pretreatment. Surprisingly, we found that emetine can trigger inflammasome activation, regardless of poly(I:C) treatment (see below). Therefore, emetine cannot be used to confirm whether stress granule formation is important for NLRP3 inflammasome activation. Consequently, we have no direct evidence suggesting that stress granule formation enhances NLRP3 inflammasome activation during the activation stage. Additionally, as pointed out by reviewer 2#, these data are largely distracting from the main point of the story. Hence, we have removed these data from the revised manuscript.

Emetine can suppress the formation of stress granules but also induce inflammation. (A) THP-1 cells were stimulated with sodium arsenite (0.5 mM) with or without emetine, fixed, and stained with an anti-G3BP1 antibody. Scale bar, 10 μ m. (B) Quantification of the percentage of cells with stress granules from 100 cells in (A). $n = 4$, mean \pm s.d., two-sided Student's t-test. (C-D) Cells with the indicated genotype were primed with LPS and treated with poly(I:C) (3 μ g/mL, 4 h), with or without emetine (10 μ g/mL, 1 h) pretreatment. Inflammation activation was measured by ELISA in (C) and immunoblot in (D).

2 NLRP3 protein level is a key limiting step for inflammasome activation (DOI: 10.3109/10409238.2012.694844, DOI: 10.1038/ncomms13727). Whether ZNFX1 could affect NLRP3 expression should be investigated.

Thank you for pointing this out. We have examined the protein levels of NLRP3 in wild-type, *ZNFX1* KO, *ZNFX1* RE, and *ZNFX1* OE cells in both THP-1 and iBMDM cell lines and did not observe significant changes in NLRP3 levels with different ZNFX1 expression levels. Additionally, overexpressing ZNFX1 in HeLa cells did not affect the protein levels of NLRP3. Therefore, ZNFX1 does not appear to affect the protein levels of NLRP3. We have included these data as Figures EV2E-H in the revised manuscript.

Results (Lines 141-145):

“Given that the NLRP3 protein level is a key limiting factor for inflammasome activation (Dowling & O’Neill, 2012; Song et al, 2016), we investigated whether ZNFX1 affects NLRP3 protein levels. Immunoblotting showed that NLRP3 protein levels were similar regardless of the ZNFX1 expression level (Fig. EV2E-H).”

3. Figure 1N showed that during LPS injection for 6 days, none of the mice in WT group (*Znfx1*+/+) has died, which is confusing. The authors should explain it.

Thank you for the comment. Different laboratories have used varying LPS concentrations (10-40 mg/kg) in LPS injection experiments⁵⁻¹², resulting in diverse mortality rates for WT mice. For instance, Hashimoto et al. reported that WT mice survived for more than 7 days with a 20 mg/kg LPS treatment⁵. However, Cho et al. reported 90% mortality after 4 days with the same concentration¹². The underlying reason for this discrepancy is unknown. It may stem from differences in the specific lots of LPS used or the condition of mice in different laboratories.

We have meticulously reviewed the original records in the lab notebook and confirmed that the concentration and protocol used were correct. However, in our experiments, the LPS concentration employed did not cause WT mice to die by the end of the experiments (we did not observe the WT mice beyond this point). It is noteworthy that the same lot of LPS caused *Znfx1*^{-/-} mice to die, suggesting that *Znfx1*^{-/-} mice are more susceptible to LPS-induced inflammation. This experiment was repeated on different days with different batches of mice (n = 4 mice for one replicate, n = 5 for the second replicate, making n = 9). Additionally, LPS induced higher levels of IL-1 β in *Znfx1*^{-/-} mice compared to WT mice (Fig. 2C). To address this concern, we have revised the text to describe the results more rigorously.

Results (Lines 162-166):

"We found that while WT mice survived for 6 days post-LPS treatment in our experimental setting (when the experiments were stopped), all Znfx1^{-/-} mice died within 4 days. This suggests that ZNFX1 protects mice from lethal inflammation (Fig. 2A-B)."

4. Figure EV 1B &G, the data showed the expression of TNF- α exist great discrepancy between LPS treatment and LPS+stimuli, which seemed not in accordance with previous study (DOI: 10.1038/ncomms13727, DOI: 10.1016/j.celrep.2024.113752). The authors should explain or further confirm this issue.

Thanks for spotting this. This discrepancy is due to a labeling issue, we mistakenly labeled the control treatment group (without LPS treatment) as the LPS treatment group. We sincerely apologize for this mistake. To confirm these results, we have replicated the experiments in revision with LPS alone control and verified the results. Fortunately, the conclusions do not change. The result is included as Appendix Figs. S2G-H in our revised manuscript.

Results (Lines 120-122):

"Notably, the ability of all the tested stimuli to induce TNF- α release was unaffected in ZNFX1 KO macrophages (Appendix Fig. S2G-H)."

5. Figure3A, the authors should perform a live cell imaging timeline of the indicated cells in order to display the location changes of NLRP3, ZNFX1 and TNG46.

We generated cells stably expressing GFP-ZNFX1, mCherry-NLRP3, and mMaroon1-TGN46. We then performed living imaging of these cells after nigericin treatment. Note, likely due to overexpression of mMaroon1-TGN46, we found more mCherry-NLRP3 colocalize with mMaroon1-TGN46 in these cells compared with cells expressing GFP-ZNFX1 and mCherry-NLRP3. This video has been included as Movie EV1 in our revised manuscript.

Results (Lines 250-253):

“Time-lapse imaging revealed that mCherry-NLRP3 translocated to dispersed TGN46+ vesicles, while GFP-ZNFX1 predominantly remained diffuse (Fig. 4A-B and movie EV1).”

Minor comments:

1. In the Abstract section, the description related stress granules or virus should be included.

Thank you for the comment. We have not included the description in the abstract since we have removed the stress granules and virus-related data in our revised manuscript.

2. The abbreviations are usually used defined at the first use in the abstract as well as in the main text, such as ZNFX1 in the abstract.

Agreed. We have defined abbreviations at the first use in the abstract as well as in the main text in our revised manuscript.

Abstract (Lines 27-29):

*“Inherited deficiency of **zinc finger NFX1-type containing 1 (ZNFX1)**, a dsRNA virus sensor, is associated with severe familial immunodeficiency, multisystem inflammatory disease, increased susceptibility to viruses, and early mortality.”*

Abstract (Lines 32-34):

*“Here, we demonstrate that ZNFX1 specifically inhibits the activation of the **NLR family pyrin domain-containing protein 3 (NLRP3)** inflammasome in response to NLRP3 activators both in vitro and in vivo.”*

Results (lines 49-53):

*“Among the various inflammasome complexes, the **NLR family pyrin domain-containing protein 3 (NLRP3)** inflammasome plays a crucial role in the host immune defence against bacterial, fungal, and viral infections (Allen et al, 2009; Kanneganti et al, 2006; Nozaki et al, 2022; Thomas et al, 2009).”*

Results (lines 83-84):

*“**Zinc finger NFX1-type containing 1 (ZNFX1)** is a highly conserved RNA helicase belonging to the SF1 helicase family.”*

3. Figure EV5I, the data could not support the corresponding conclusion well (line 295).

Thank you for pointing this out. I believe the reviewer is referring to the fact that MG132, CQ, and 3-MA pretreatment had no effect on protein levels in our original submission. This discrepancy may be due to the short pretreatment time of each chemical (1 hour). To address the reviewer's concern, we subjected the cells to 4 hours of pretreatment with MG132, CQ, and 3-MA. The following results suggest that 4 hours of treatment with MG132, CQ, or 3-MA inhibits the proteasome or autophagy:

First, the protein levels of several proteins, such as ZNFX1 and NLRP3, increased after MG132 treatment, consistent with MG132's role in inhibiting proteasome-mediated protein degradation; Second, P62 decreased after NLRP3 inflammasome activation, but this decrease was blocked by CQ or 3-MA treatment; Third, the levels of IL-1 β and caspase-1 P20 increased after MG132, CQ, or 3-MA treatment compared with DMSO control treatment, consistent with previous reports^{13, 14}. However, the ZNFX1 protein level decreased after NLRP3 inflammasome activation regardless of proteasome or autophagy inhibitors pretreatment. The results are included as Fig. EV4G-H in our revised manuscript.

Results (lines 323-328):

"We found that the protein levels of ZNFX1, NLRP3, IL-1 β , and caspase-1 P20 were increased upon MG132, CQ, or 3-MA treatment, as expected and consistent with previous reports (Han et al, 2019; Song et al., 2016). Remarkably, the level of full-length ZNFX1 significantly decreased regardless of the inhibitor that was administered (Fig. EV4G-H)."

4. The expression of GAPDH or Tubulin is too weak, such as Figure 4A, C.

We have increased the brightness of the GAPDH and Tubulin images to an appropriate level in our revised manuscript. Figures 4A and 4C are now Figures 5A and 5C in the revised manuscript.

5. There are numerous grammatical mistakes in the manuscript (such as Line197).

We apologize for the grammatical mistakes in our first submission. We have carefully checked our manuscript using ChatGPT and Grammarly to identify and correct any grammatical errors. Additionally, we utilized Springer Nature Author Services to polish the manuscript. We hope you would agree that the clarity and quality of our revised manuscript has been improved, with grammatical mistakes eliminated.

References cited in Response to Reviewers:

1. Wang, Y. *et al.* Mitochondria-localised ZNFX1 functions as a dsRNA sensor to initiate antiviral responses through MAVS. *Nat Cell Biol* **21**, 1346-1356 (2019).
2. Le Voyer, T. *et al.* Inherited deficiency of stress granule ZNFX1 in patients with monocytosis and mycobacterial disease. *Proc Natl Acad Sci U S A* **118** (2021).
3. Kedersha, N. *et al.* Stress granules and processing bodies are dynamically linked sites of mRNP remodeling. *J Cell Biol* **169**, 871-884 (2005).
4. Bussi, C. *et al.* Stress granules plug and stabilize damaged endolysosomal membranes. *Nature* **623**, 1062-1069 (2023).
5. Wu, J. *et al.* Mik1 knockout mice demonstrate the indispensable role of Mik1 in necroptosis. *Cell Res* **23**, 994-1006 (2013).
6. Lew, W.Y. *et al.* Recurrent exposure to subclinical lipopolysaccharide increases mortality and induces cardiac fibrosis in mice. *PLoS One* **8**, e61057 (2013).
7. Hashimoto, R., Koide, H. & Katoh, Y. MEK inhibitors increase the mortality rate in mice with LPS-induced inflammation through IL-12-NO signaling. *Cell Death Discov* **9**, 374 (2023).
8. An, N. *et al.* Pretreatment of mice with rifampicin prolongs survival of endotoxic shock by modulating the levels of inflammatory cytokines. *Immunopharmacol Immunotoxicol* **30**, 437-446 (2008).
9. Liu, S. *et al.* TNFR2 expression on non-bone marrow-derived cells is crucial for lipopolysaccharide-induced septic shock and downregulation of soluble TNFR2 level in serum. *Cell Mol Immunol* **8**, 164-171 (2011).
10. Aksoy, E. *et al.* The p110delta isoform of the kinase PI(3)K controls the subcellular compartmentalization of TLR4 signaling and protects from endotoxic shock. *Nat Immunol* **13**, 1045-1054 (2012).
11. Jang, J.C. *et al.* Human resistin protects against endotoxic shock by blocking LPS-TLR4 interaction. *Proc Natl Acad Sci U S A* **114**, E10399-E10408 (2017).
12. Cho, E.J. *et al.* Zwitterionic chitosan for the systemic treatment of sepsis. *Sci Rep* **6**, 29739 (2016).
13. Moscat, J. & Diaz-Meco, M.T. p62 at the crossroads of autophagy, apoptosis, and cancer. *Cell* **137**, 1001-1004 (2009).
14. Shi, C.S. *et al.* Activation of autophagy by inflammatory signals limits IL-1beta production by targeting ubiquitinated inflammasomes for destruction. *Nat Immunol* **13**, 255-263 (2012).

Dear Prof. Wan,

Thank you for the submission of your revised manuscript to The EMBO Journal and your patience during its peer review. It has now been seen by two of the original referees who previously assessed the earlier version of your manuscript, and we have received their comments (included below). As you will see, the referees acknowledge that all previously raised concerns have been addressed and the manuscript has been improved according to their suggestions.

While referee #3 has no further comments, referee #2 has a few more suggestions for minor changes in the presentation of some results and explanation/clarification of the used methods. We kindly ask you to fully address these remaining points in a final version of your manuscript. Please also include in your resubmission a point-by-point response to these comments, explaining in detail any new changes to the manuscript.

From the editorial side, there are also a few changes and corrections that we need from you before we can proceed with acceptance of your manuscript for publication:

- Please include a list of up to 5 general keywords after the Abstract of your revised manuscript.
- The Materials and Methods need to be described in the manuscript using our "Structured Methods" format, which is now required for all research articles. According to this format, the Materials and Methods section includes a single "Reagents and Tools Table" -listing key reagents, experimental models, software and relevant equipment and including their sources and relevant identifiers- followed by a "Methods and Protocols" section describing the methods using a step-by-step protocol format. The aim is to facilitate adoption of the methodologies across labs. More information on this format as well as detailed instructions, examples, and a template (.docx) for the "Reagents and Tools Table" can be found in our author guide: <https://www.embopress.org/page/journal/14602075/authorguide#structuredmethods>.
- Please include in your revised Materials and Methods the details of the authority granting ethics approval of the experiments involving mice, including the reference number for approval.
- Please change the heading of your conflict-of-interest statement to "Disclosure and competing interests statement".
- The author contributions statement should be removed from the manuscript file. Instead, we now use CRediT to specify the contributions of each author in the journal submission system. Please feel free to use the free text box to provide more detailed descriptions during submission. See also our guide to authors for more information: <https://www.embopress.org/page/journal/14602075/authorguide#authorshippinguidelines>.
- We noticed that there are no callouts in the manuscript for Appendix Table S1. Please make sure that all Appendix items are called out in the manuscript as appropriate.
- Please move the Contents table of your Appendix to its first (title) page.
- Please ZIP together all Source Data for EV items and Appendix Figures in a master ZIP folder called "EV Appendix items Source Data".
- Please note that EMBO press papers are accompanied online by:
 - A) a short (2 sentences) summary of the findings and their significance,
 - B) 2-5 short bullet points highlighting the key results, and
 - C) a synopsis image in .jpg or .png format that is exactly 550 pixels wide and 300-600 pixels high (the height is variable). Please note that the text needs to be legible at the final size. Please upload this information along with your revised manuscript (the text for A and B should be provided in a separate Word file).
- During a standard image check, we detected the following potential aberrations in your Figure set that must be addressed before we can proceed with acceptance of the manuscript:
 1. Cell image re-use between Figure 4C and EV5C (WT control). The re-use is not mentioned in the figure legend. If the re-use in this case is intentional and experimentally justified, it should be clearly disclosed in the figure legends.
 2. Blot re-use (Tubulin control) between Figure 6E and Figure EV4G. The re-use is not described in the figure legend. Source data for the two Figures are also the same. If the re-use in this case is intentional and experimentally justified, it should be clearly disclosed in the figure legends.Please check both issues carefully, explain/clarify in your cover letter or point-by-point response, and describe any changes to the Figures and/or their legends.
- Please rename your Movie files to "Movie EV1" to "Movie EV4", and also update their callouts throughout the manuscript accordingly. For each Movie, its legend should be zipped together with the Movie File in a Word/text file.

- Please note that the exact p values are not provided in the legend of Figure 2a.
- Please indicate the statistical test used for data analysis in the legends of Figures 2a, c-d; 7h; EV 1b-k; EV 2a-c, i-l, n; EV 4e-f, h-k; EV 5b, f.
- Please note that information related to "n" is missing in the legends of Figures 2c-d; 7h; EV 4e-f, i-k; EV 5b, f.
- Although "n" is provided, please describe the nature of entity for "n" in the legend of Figure 1f.
- Please note that the error bars are not defined in the legends of Figures 2c-f; 7h; EV 4e-f, i-k; EV 5b, f.
- Please note that the scale bar needs to be defined for Figures 2b, g.

Please also note that as part of the EMBO publications' Transparent Editorial Process, The EMBO Journal publishes online a Peer Review File along with each accepted manuscript. This File will be published in conjunction with your paper and will include the referee reports, your point-by-point response and all pertinent correspondence relating to the manuscript. You can opt out of this by letting the editorial office know (contact@embojournal.org). If you do opt out, the Peer Review File link will point to the following statement: "No Peer Review File is available with this article, as the authors have chosen not to make the review process public in this case."

We look forward to seeing a final version of your manuscript as soon as possible. Please use this link to submit your revision:
<https://emboj.msubmit.net/cgi-bin/main.plex>

Best regards,

Ioannis

Referee #2:

The authors have responded to all my concerns. They have improved data quantification and have deleted distracting data re: G3BP and stress granules. I only have minor presentation suggestions.

The authors now provide graphs in supplementary figure EV1 showing the results of three biological replicates to support the findings shown in Figure 1 that ZNFX1 knock out cells show increased response to various inflammation agonists. The results are presented as the ratio of IL-beta or Casp-1 to tubulin. These ratios have been normalized to 1 for WT cells, but this is not indicated in the figure legend. Also it appears that each WT replicate has been normalized to 1. The authors should consider normalizing the AVERAGE of the three WT replicates to 1 so the variability between WT replicates can be shown. Throughout the paper the authors should describe in more detail how they set up their biological and technical replicates (were each experimental replicate paired to one wild-type control?), they should also provide the raw data in excel file format, and the formulas used to generate the values shown in for each graph.

Finally, they should indicate in the legends that white bars indicate WT cells and red bars indicate ZNFX1 knock out cells - this was not always clear for example in Fig. 1

Referee #3:

The authors have addressed all my concerns. No further comments.

Dear Ioannis,

Our response is in Blue.

Referee #2:

The authors have responded to all my concerns. They have improved data quantification and have deleted distracting data re: G3BP and stress granules. I only have minor presentation suggestions.

The authors now provide graphs in supplementary figure EV1 showing the results of three biological replicates to support the findings shown in Figure 1 that ZNFX1 knock out cells show increased response to various inflammation agonists. The results are presented as the ratio of IL-beta or Casp-1 to tubulin. These ratios have been normalized to 1 for WT cells, but this is not indicated in the figure legend. Also it appears that each WT replicate has been normalized to 1. The authors should consider normalizing the AVERAGE of the three WT replicates to 1 so the variability between WT replicates can be shown. Throughout the paper the authors should describe in more detail how they set up their biological and technical replicates (were each experimental replicate paired to one wild-type control?), they should also provide the raw data in excel file format, and the formulas used to generate the values shown in for each graph.

We thank the reviewer for the great suggestion. For each western blot experiment, we calculated the ratio of IL- \$\beta\$ or Casp-1 to tubulin relative to the corresponding WT control. Since we conducted three biological replicates, we had three WT controls. Each WT control was arbitrarily set to 1, allowing the ratio of each experimental condition to be compared with its respective WT control within the same biological replicate. This approach is commonly used in published studies^{1,2}. We have included the following explanation in the figure legends for Figures 5E-F, 5H, 6G-I, 7C, EV1B-K, EV2F, EV2I-L, EV4H, and Appendix S3D.

“For each biological replicate, band intensity was measured using ImageJ. The WT control was set to 1, and the ratio for ZNFX1 KO cells was calculated by dividing their intensity by the corresponding WT control intensity.”

We have provided the raw data in Excel format, along with the formulas used in the Excel files. These are available in the source data files.

Finally, they should indicate in the legends that white bars indicate WT cells and red bars indicate ZNFX1 knock out cells - this was not always clear for example in Fig. 1

Thank you for pointing this out. We believe the reviewer is referring to the missing information regarding the white and red bars in some figures. We have now included bars to indicate that white bars represent WT cells and red bars represent ZNFX1 knock-out cells in Figures EV1B-K, EV2I-L, and EV2N.

Referee #3:

The authors have addressed all my concerns. No further comments.

Thank you for dedicating your time and effort to review our manuscript. We sincerely appreciate your valuable insights and suggestions, which have greatly contributed to enhancing the quality of our study.

References for point-to-point response:

1. Liu, T. et al. USP19 suppresses inflammation and promotes M2-like macrophage polarization by manipulating NLRP3 function via autophagy. *Cell Mol Immunol* **18**, 2431-2442 (2021).
2. Yu, H. et al. Global crotonylome reveals CDYL-regulated RPA1 crotonylation in homologous recombination-mediated DNA repair. *Sci Adv* **6**, eaay4697 (2020).

Dear Prof. Wan,

Congratulations on an excellent manuscript, I am very pleased to inform you that it has been accepted for publication in The EMBO Journal. Thank you very much for your comprehensive responses to the referee concerns and for addressing all editorial requests.

Your manuscript will now be processed for publication by EMBO Press. It will be copy edited and you will receive page proofs prior to publication. Please note that you will be contacted by Springer Nature Author Services to complete licensing and payment information.

If you have any questions, please do not hesitate to contact the Editorial Office. Thank you for your contribution to The EMBO Journal. Working with you has been a pleasure!

Best wishes,

Ioannis
